# REVISIT WHAT YOU SEE: DISCLOSE LANGUAGE PRIOR IN VISION TOKENS FOR LVLM DECODING

## ABSTRACT

Large Vision–Language Models (LVLMs) achieve strong performance across multimodal tasks by integrating visual perception with language understanding. However, how vision information contributes to the model's decoding process remains under-explored, as reflected in frequent hallucinations. Through a series of analyses, we found that (i) vision tokens provide meaningful visual information even when when hallucinations occur, and (ii) their semantics are encoded in the textual space and become explicit under appropriate vocabulary constraints. Building on these observations, we propose ReVisiT, a simple training-free decoding method that references vision tokens to guide text generation. Our approach leverages the semantic information embedded within vision tokens by projecting them into the text token distribution. Specifically, ReVisiT dynamically selects the most relevant vision token at each decoding step via context-aware constrained divergence minimization, and using its constrained projection to refine the output distribution to better incorporate visual semantics. Across five benchmarks on recent LVLMs, ReVisiT consistently enhances visual grounding with minimal computational overhead, and achieves competitive or superior results to state-of-the-art decoding baselines while reducing computational cost by up to $2\times$.

## 1 INTRODUCTION

With the recent success of Large Language Models (LLMs) (Touvron et al., 2023; Achiam et al., 2023; Team et al., 2023), Large Vision-Language Models (LVLMs) have emerged as powerful multimodal architectures that integrate visual perception with language understanding and text generation (Dai et al., 2023; Liu et al., 2024a; Zhu et al., 2024; Bai et al., 2025; Zhu et al., 2025). Typically, LVLMs encode visual inputs into the LLM decoder's text embedding space as *vision tokens*, which are then processed alongside text tokens during decoding. This way of treating vision tokens as static auxiliary contexts, similar to retrieval-augmented generation (Lewis et al., 2020; Gao et al., 2023) in LLMs, enables the construction of complex multimodal systems in a relatively simple manner. However, this approach is often insufficient to capture the unique characteristic and role of vision tokens as sole carriers of visual information, as denoted by hallucinations frequently observed in LVLM's text outputs (Leng et al., 2024; Favero et al., 2024; Huo et al., 2025; Rohrbach et al., 2018; Li et al., 2023b; Guan et al., 2024). To mitigate this, research has begun to expand the understanding of vision tokens along with their additional utilization (Jiang et al., 2025a;b). Yet, it is still under-explored *how vision tokens influence the decoding process of LVLMs* and *what kinds of textual semantics they encode*.

To investigate this direction, we conduct a series of analyses on vision tokens, leading to two key insights. First, by examining the output distributions of LVLMs particularly at hallucinated steps, we find that ground-truth objects often remain among the model's high-probability candidates and this visual grounding signal in output distribution indeed comes from the vision tokens. Second, we interpret the semantics of vision tokens by projecting them into the text token distribution using the LLM decoder's language modeling head, as both are embedded in the same space. Here, we observe that the textual semantics of vision tokens is initially obscured, but the interpretable and well-aligned semantics could be discovered in the projected distribution with the proper constraints to focus on specific subset. Overall, these findings suggest that vision tokens intrinsically encode object-level semantics, though such signals typically remain latent under conventional unconstrained decoding.

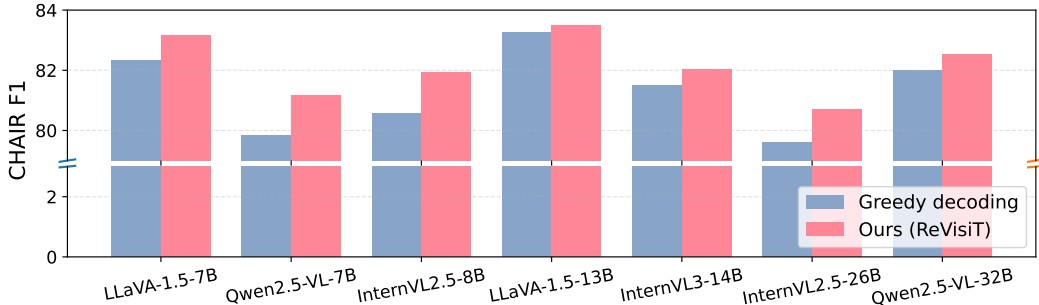

Figure 1: **ReVisiT across various model sizes and architectures.** We evaluate on the CHAIR benchmark and report F1. Consistent improvements hold across different size buckets (7–8B / 13–14B / 26–32B) and architectures, demonstrating scalability and model-agnostic effectiveness. Full results are provided in Table 9.

Building on this observation, we propose ReVisiT, a simple yet effective decoding strategy that **Re**ferencing **Visi**on **T**okens to guide the text generation process of LVLMs. At each step, ReVisiT selects the most relevant vision token based on *context-aware constrained divergence* from the current output distribution, and uses its constrained projection as reference to refine token-level logits. This activates the intrinsic semantics of vision tokens without requiring additional training, architectural modifications, external modules, or multi-pass inference.

As highlighted in Fig. 1, ReVisiT consistently improves performance across six model sizes spanning three architecture, demonstrating scalability and model-agnostic effectiveness. To be specific, on three hallucination benchmarks—HallusionBench (Guan et al., 2024), CHAIR (Rohrbach et al., 2018), and POPE (Li et al., 2023b)—ReVisiT reduces hallucinations robustly on both LLaVA-1.5 (Liu et al., 2024a) and Qwen2.5-VL (Bai et al., 2025). Beyond hallucination, we further validate its effectiveness on general-purpose benchmarks, including VQAv2 (Antol et al., 2015) and MMMU (Yue et al., 2024), indicating that the semantics surfaced by our context-aware constrained projection generalize across tasks and architectures.

In summary, our contributions are as follows:

○ We provide quantitative and qualitative evidences that vision tokens intrinsically encode interpretable semantics exposed when projected over a *semantically coherent vocabulary subset*.

○ We introduce ReVisiT, a simple yet effective LVLM decoding method that **Re**ferencing **Visi**on **T**okens via *context-aware constrained divergence*.

○ We validate ReVisiT across *six model sizes spanning three architectures*, reducing hallucinations and yielding gains on five benchmarks, while maintaining inference cost.

## 2 MOTIVATION

In this section, we investigate whether vision tokens encode interpretable semantics in LVLM's text token space that can directly inform its decoding process. To conduct systematic analyses, we adopt the CHAIR benchmark (Rohrbach et al., 2018), as it provides a principled definition of hallucinations based on 80 MSCOCO (Lin et al., 2014) object categories with 403 synonyms in total; for each sample, the set of synonyms of annotated object categories is regarded as ground-truth (GT) objects, while the remaining category synonyms are regarded as hallucinated (Hal) objects. We refer to a *hallucinated step* as a decoding step where generated token corresponds to an object in this MSCOCO synonym set but is not part of the ground-truth annotations for the given image.

Building on this definition, we sample 500 images from the MSCOCO validation set and generate captions for all images. All quantitative analyses are performed with *Qwen2.5-VL-7B* (Bai et al., 2025) using vanilla greedy decoding with a maximum of 512 tokens, prompted with `"Please describe this image in detail."`, which is a standard input prompt for CHAIR.

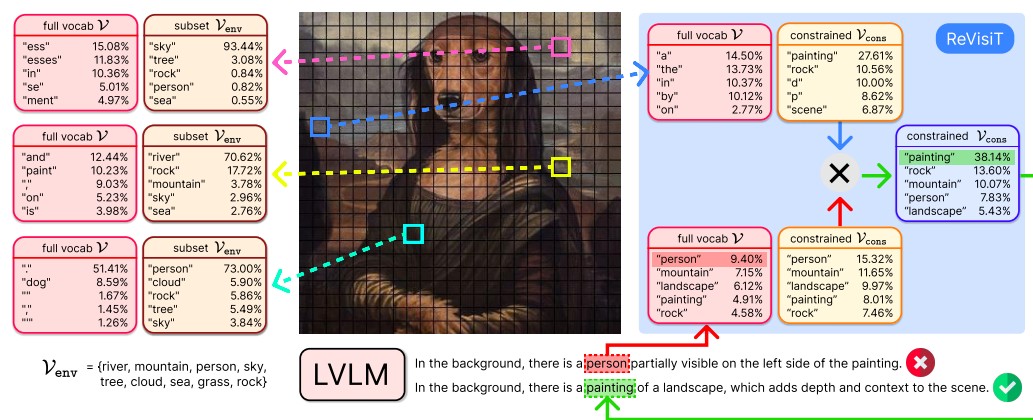

Figure 2: **Motivation.** We qualitatively analyzed various vision tokens with LLaVA-1.5-7B (Liu et al., 2024a). Dotted arrows represent vision token projection over specified vocabulary set. For each box, representing text token distribution, we annotated top-5 probable text tokens. Left part illustrate the effectiveness of vocabulary constraint, whereas right part shows the distribution shift during ReVisiT. See Appendix C.1 for a detailed discussion of the underlying values and analysis.

## 2.1 HOW DO VISION TOKENS AFFECT THE OUTPUT DISTRIBUTION OF LVLM?

To examine how vision tokens affect the decoding step of LVLMs, we first ask whether ground-truth (GT) objects remain among the model's top-probability predictions at hallucinated steps. We use a *at-least-one recall@*k defined on a probability distribution over the vocabulary: for a given distribution, we measure whether at least one GT object synonym appears within its top-$k$ predictions. Among the 500 generated captions, 176 captions contain at least one hallucinated step, and we obtain 190 hallucinated steps for step-level analysis. For each hallucinated step, we report *at-least-one recall@*k on the model's output distribution at that step, excluding the greedy (top-1) prediction from the candidate set (details in Appendix A.1.2).

The results are summarized in Fig. 3. It is observable that GT objects frequently remain accessible in output distribution: in 63.2% of hallucinated steps, at least one GT object appears within the top-50 candidates, and this recall rises to 95.8% within the top-500. Given the vocabulary size of 151,665 tokens, top-500 corresponds to only 0.33% of the full vocabulary. A qualitative example is also presented in Fig. 2 (right), where the model incorrectly selects "person" as the greedy prediction, yet visually grounded alternatives such as "mountain," "landscape," and "painting" appear among the top-5 candidates. This indicates that relevant visual candidates are assigned high probabilities in the output distribution, showing that the LVLM indeed possesses the correct visual information even when hallucinations occur. The 95% confidence intervals are reported in Appendix A.1.2.

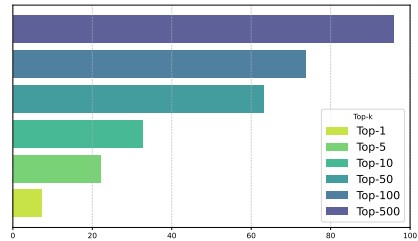

Figure 3: **Ground-truth objects frequently remain in top-probability predictions.** At 190 hallucinated steps, at least one GT object is recalled in 95.8% of cases within the top-500 predictions, corresponding to only 0.33% of full vocabulary (151,665 tokens).

We further conduct a fine-grained comparison between the occurrences of GT and hallucinated object subsets during the decoding. Specifically, for each generated token, we track the maximum probability among objects in each subset. Then, we report the average of these per-step maxima as *GT Max Avg* for GT object subsets and *Hal Max Avg* for hallucinated object subsets, respectively (see detailed definition in Appendix A.1.3). To investigate the contribution of vision tokens by isolating them, we additionally report *w/o image* as an image-absent setting where we remove all vision tokens and perform teacher forcing with the previously generated caption as input while keeping decoding otherwise identical.

As shown in Table 1, with vision tokens present, GT objects receive higher probability mass than hallucinated ones on average (GT > Hal); removing the image tokens reverses the ordering (Hal > GT), confirming a grounding contribution from visual input. At hallucinated steps, the reversed ordering (Hal > GT) persists even with the presentation of image, indicating that *language priors* (*i.e.*, textual co-occurrence biases from pre-training (Leng et al., 2024; Favero et al., 2024)) can dominate the prediction at specific positions. We further validate this ordering shift using a paired nonparametric Wilcoxon signed-rank test (Wilcoxon, 1945),

Table 1: **Comparison of Hal Max Avg and GT Max Avg.** The results for (i) all output tokens across all 500 CHAIR samples (88,706 tokens), (ii) all output tokens within hallucinated samples (176 samples, 35,116 tokens), and (iii) hallucinated decoding steps within hallucinated samples (176 samples, 190 steps). The higher value between Hal and GT is highlighted in **bold**.

| Analysis Setting | Condition | GT Max Avg | Hal Max Avg |
|---|---|---|---|
| All tokens | w/ image | **24.49** | 20.24 |
| | w/o image | 21.83 | **22.66** |
| All tokens | w/ image | **24.42** | 21.74 |
| (in hallucinated samples) | w/o image | 21.73 | **23.88** |
| Hallucinated tokens | w/ image | 32.00 | **36.44** |
| | w/o image | 30.26 | **35.23** |

which compares the signs and magnitudes of paired step-level differences without assuming normality, showing a significant shift toward GT when moving from w/o-image to w/-image condition in the full-token analyses (Table 1, settings (i) and (ii)), while no significant shift is observed on the hallucinated-step subset (See Appendix A.1.3 for detailed discussion).

## 2.2 WHAT TEXTUAL SEMANTICS ARE ENCODED IN VISION TOKENS?

To identify the source of visual grounding signals, we focus on vision tokens, the sole carriers of visual information in LVLMs. Vision tokens share the same embedding space with text tokens and are autoregressively processed during decoding like them. This motivates probing their representations by projecting them through the model's language-modeling head to obtain the text token distribution over the vocabulary, analogous to logit-lens and early-exit analyses for text token embeddings (nostalgebraist, 2020; Chuang et al., 2024; Wang et al., 2025). Specifically, for each of 500 MSCOCO validation images, we project each vision token's final-layer hidden state to obtain the text token distribution, yielding 173,173 vision token projections. (see details in Appendix A.1.4).

First, to assess the presence of visual grounding signals, we compute token-level *at-least-one recall*@k for each vision token projection, following the definition in Sec. 2.1. Quantitatively, across 173,173 vision tokens, *at-least-one recall*@k is low under the full vocabulary: only **2.03%** of projections surface a GT synonym at Top-1, increasing to **21.72%** at Top-30. Qualitative inspection also indicates that tokens corresponding to background regions often yield generic function words, with object labels rarely appearing (red boxes on the left side of Fig. 2). These observations indicate that, under the full vocabulary, vision-token projections are diffuse and difficult to interpret.

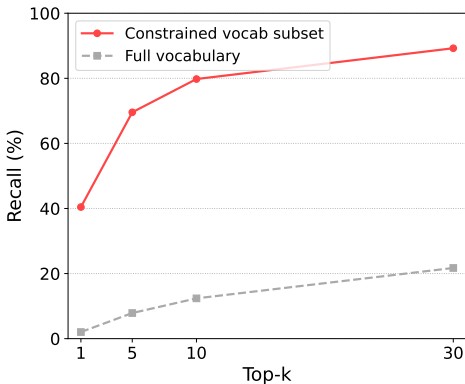

Figure 4: **Vision token projection.** Vision tokens reveal rich semantics when projected over semantically coherent subsets.

In contrast, constraining the projection to *semantically coherent* vocabulary subsets reveals substantially clearer alignment with visual semantics. When restricting the prediction space to the CHAIR-derived object subset (176 single-token synonyms covering 62 categories), *at-least-one recall*@k on the *same* 173,173 projections increases sharply to **40.44%** at Top-1—about $20\times$ higher than the naïve full-vocabulary projection (2.03%). At Top-30, recall reaches **89.24%** while inspecting only $30/176 \approx 17\%$ of the subset. For comparability, both settings evaluate GT synonyms from the same 62 categories. Moreover, when we manually define domain-specific subsets, such as environment-related terms including "river," "mountain," and "tree", the projected distribution aligns with visually salient regions and reveals object labels that remain hidden under the full-vocabulary setting (yellow boxes on the left side of Fig. 2).

These findings provide both quantitative and qualitative evidence that vision tokens intrinsically encode object-level semantics, although such signals are not easily observable without appropriate semantic constraints. Details are presented in Appendix A.1.4.

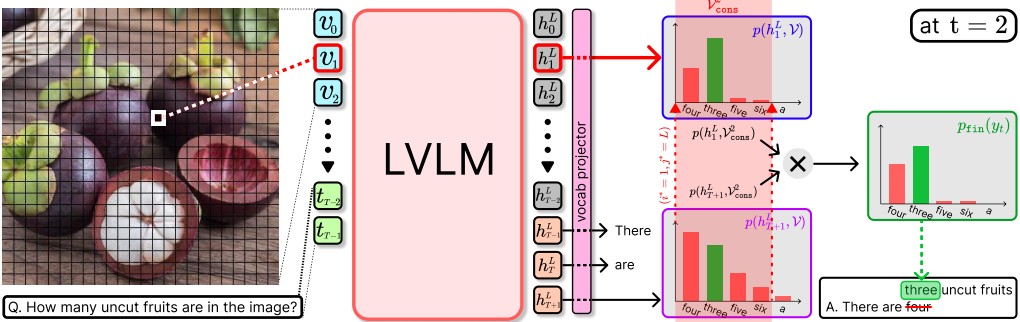

Figure 5: **An overview of ReVisiT.** At each decoding step, ReVisiT (1) constrains the vocabulary $\mathcal{V}$ to $\mathcal{V}^t_{\text{cons}}$, (2) projects vision token embeddings into $\mathcal{V}^t_{\text{cons}}$ and selects most relevant token, and (3) refines the final output distribution. ReVisiT leverages vision tokens to serve as reference signals for decoding, enhancing visual grounding without additional forward passes or external supervision.

## 3  REVISIT: REFERENCING VISION TOKENS TO GUIDE TEXT GENERATION

Based on insights from Sec. 2, we propose ReVisiT, a decoding method that adaptively leverages intermediate vision token representations as internal reference signals to guide text generation, enhancing visual grounding with minimal computational overhead. At each decoding step, ReVisiT focuses the decoding process onto a context-relevant vocabulary subset and refines the output by referencing the most relevant vision token. To be specific, ReVisiT consists of the following three main steps: (1) adaptively constraining the vocabulary based on the vanilla output distribution, (2) projecting and selecting vision token hidden states within the subset, and (3) refining the output distribution through element-wise combination and normalization to generate the next token.

**Preliminary.** Let denote input text tokens (*e.g.,* query) and vision tokens as $\mathbf{x}$ and $\mathbf{v}$, respectively. Specifically, the vision tokens $\mathbf{v}$ are generated from the given input image, by forwarding it into vision encoder and cross-modality projector sequentially (Liu et al., 2024a; Bai et al., 2025). Then, our goal is to generate output text tokens $\mathbf{y}$ using LVLM $\mathcal{M}$; following the auto-regressive nature of LLM, output tokens $\mathbf{y}$ (*e.g.,* response) are sequentially generated, *i.e.,* $y_t \sim \mathcal{M}(\cdot|\mathbf{v}, \mathbf{x}, \mathbf{y}_{<t-1})$. Let assume that LVLM $\mathcal{M}$ has $L$ decoding layers and text token vocabulary $\mathcal{V}$, and denote its hidden state of $l$-th layer as $h^l$. The output distribution is parameterized by model weights $\theta$ and denoted as $p_\theta$. Then, at time step $t$, the output probability over $\mathcal{V}$ is calculated as below:

$$p_\theta(h^L_{T+t-1}, \mathcal{V}) = \text{softmax}_\mathcal{V}\left(\phi(h^L_{T+t-1})\right), \quad \sum_{w \in \mathcal{V}} p_\theta(w|h^L_{T+t-1}, \mathcal{V}) = 1, \quad (1)$$

where $T = |\mathbf{v}| + |\mathbf{x}|$, $\phi$ is the language modeling head which maps final hidden state of LVLM into logits over vocabulary $\mathcal{V}$, and $\text{softmax}_\mathcal{V}$ indicates a softmax operation over $\mathcal{V}$. Then, the output token $y_t$ is sampled from the obtained distribution:

$$y_t \sim p_\theta(h^L_{T+t-1}, \mathcal{V}). \quad (2)$$

### 3.1  ADAPTIVE CONTEXT-AWARE VOCABULARY SUBSET CONSTRUCTION

At each decoding timestep $t$, LVLM $\mathcal{M}$ generate output distributions $p_\theta(h^L_{T+t-1}, \mathcal{V})$ over the entire vocabulary space $\mathcal{V}$. However, as observed in Figure 2, the original distribution is often diffusely spread across many irrelevant tokens. Therefore, we dynamically constrain a vocabulary $\mathcal{V}$ into subset $\mathcal{V}^t_{\text{cons}}$ more suitable to capture the plausible semantic space of the output distribution. By restricting the output space to a smaller, contextually focused subset, this constraint reduce semantic distraction and facilitate more precise integration of visual information. To be specific, following Li et al. (2023a), we adaptively define the vocabulary subset $\mathcal{V}^t_{\text{cons}}$ as below:

$$\mathcal{V}^t_{\text{cons}} = \left\{w \in \mathcal{V} : p_\theta(w|h^L_{T+t-1}, \mathcal{V}) \geq \alpha \cdot \max_{w'} p_\theta(w'|h^L_{T+t-1}, \mathcal{V})\right\}, \quad (3)$$

where $\alpha \in (0, 1)$ controls the sparsity of the selected candidates. This vocabulary subset $\mathcal{V}_{\text{cons}}^t$ serves as a dynamically adapted output space, which all subsequent vision token projections and refinements are performed to enhance visual grounding without introducing additional computation.

## 3.2 Vision token projection and selection over subset

Given the dynamically constrained vocabulary subset $\mathcal{V}_{\text{cons}}^t$ at each decoding timestep $t$, we next describe how vision token hidden states $h_i^j$ are projected and selected within this constrained space. Let $\mathcal{J}$ as the set of candidate decoder layers where vision token hidden states are selected. Specifically, for each $h_i^j$, where $i$ denotes the index of the vision tokens and $j$ denotes the candidate layer, we obtain the projected distribution over the constrained vocabulary $\mathcal{V}_{\text{cons}}^t$:

$$p_\theta(h_i^j, \mathcal{V}_{\text{cons}}^t), \quad \text{for} \quad i \in \{0, \dots, |\mathbf{v}| - 1\}, \; j \in \mathcal{J}. \tag{4}$$

Intuitively, this distribution represents the likelihood of each candidate token within $\mathcal{V}_{\text{cons}}^t$ conditioned on the vision token embedding $h_i^j$. Next, to identify the vision token most relevant to the current decoding context, we compare each vision token distribution $p_\theta(h_i^j, \mathcal{V}_{\text{cons}}^t)$ against the vanilla output distribution $p_\theta(h_{T+t-1}^L, \mathcal{V}_{\text{cons}}^t)$ by computing the Jensen-Shannon Divergence (JSD):

$$(i^*, j^*) = \arg\min_{i,j} \text{JSD}\left(p_\theta(h_{T+t-1}^L, \mathcal{V}_{\text{cons}}^t) \,\|\, p_\theta(h_i^j, \mathcal{V}_{\text{cons}}^t)\right). \tag{5}$$

Here, $i^*$ and $j^*$ denote the indices of the selected vision token and decoder layer, respectively. We adopt JSD as the divergence metric due to its symmetric properties and bounded output range.

**Remark.** For the efficient implementation, the vision token projections over the full vocabulary $\mathcal{V}$, $p_\theta(h_i^j, \mathcal{V})$, are precomputed once before decoding begins. Then, at each timestep $t$, we apply slicing and masking to adapt the cached projections to the dynamically constrained subset $\mathcal{V}_{\text{cons}}^t$, enabling efficient selection without additional forward computation.

## 3.3 Output logit refinement using distribution from selected vision token

After identifying the most contextually relevant vision token at each decoding step, we then refine the vanilla output distribution guided by the reference visual grounding signal from the selected vision token. Specifically, given the vanilla output distribution $p_\theta(h_{T+t-1}^L, \mathcal{V}_{\text{cons}}^t)$ and the selected vision token distribution $p_\theta(h_{i^*}^{j^*}, \mathcal{V}_{\text{cons}}^t)$, we combine them through element-wise multiplication:

$$p_{\text{fin}}(y_t) \propto \begin{cases} p_\theta(y_t|h_{T+t-1}^L, \mathcal{V}_{\text{cons}}^t) \times p_\theta(y_t|h_{i^*}^{j^*}, \mathcal{V}_{\text{cons}}^t), & \text{if } y_t \in \mathcal{V}_{\text{cons}}^t, \\ 0, & \text{otherwise.} \end{cases} \tag{6}$$

The combined distribution $p_{\text{fin}}$ is subsequently normalized over $\mathcal{V}_{\text{cons}}^t$ to form a valid probability distribution, and the next output token $y_t$ is sampled from the normalized $p_{\text{fin}}$. This refinement mechanism allows the decoding process to dynamically integrate visual grounding signals extracted from vision tokens, without introducing additional inference passes or architectural modifications. The overall algorithm of ReVisiT is presented in Algorithm 1.

# 4 Experiments

## 4.1 Setups

**Models and baselines.** To validate the generality of our method across different architectures and capabilities, we conduct experiments using two representative LVLMs: *LLaVA-1.5-7B* (Liu et al., 2024a) as a widely adopted conventional baseline, and *Qwen2.5-VL-7B* (Bai et al., 2025) as a representative of state-of-the-art. As baselines, we adopt various intra-alignment decoding strategies to compare the effectiveness of our method in enhancing visual grounding. (a) *Greedy decoding* is a standard decoding baseline, and (b) *DoLa* (Chuang et al., 2024) improves factuality by contrasting output logits from earlier and later decoder layers at each timestep, and has been widely adopted for hallucination mitigation in LVLMs despite being originally proposed for LLMs. For intra-alignment

Table 2: **HallusionBench**. We report accuracies (%) on five metrics: *qAcc* (Question-pair accuracy), *fAcc* (Figure accuracy), *Easy aAcc*, *Hard aAcc*, and *aAcc* (All accuracy). Higher scores (↑) indicate better performance for all metrics. he best results in each setting are **bolded**, and the second-best are underlined

| Method | LLaVA-1.5-7B | | | | | Qwen2.5-VL-7B | | | | |
|---|---|---|---|---|---|---|---|---|---|---|
| | qAcc ↑ | fAcc ↑ | Easy aAcc ↑ | Hard aAcc ↑ | aAcc ↑ | qAcc ↑ | fAcc ↑ | Easy aAcc ↑ | Hard aAcc ↑ | aAcc ↑ |
| Greedy | 11.55 | 22.61 | 44.40 | 48.54 | 48.22 | 29.96 | 34.78 | **65.70** | 53.65 | 61.08 |
| DoLA | 13.36 | 22.61 | **45.85** | 49.27 | 49.24 | 27.08 | 30.87 | 62.09 | 51.46 | 58.04 |
| VCD | 14.44 | 22.61 | 44.77 | 47.45 | 47.55 | 29.60 | 35.22 | 64.62 | 54.74 | 61.25 |
| M3ID | 11.91 | 23.04 | 43.68 | 49.27 | 48.22 | 30.32 | 33.48 | 62.82 | 54.38 | 60.07 |
| CODE | 10.47 | 20.43 | 41.52 | 47.81 | 46.36 | 28.88 | 31.74 | 57.76 | 54.74 | 57.53 |
| SID | 12.27 | 21.30 | 44.77 | 48.18 | 48.22 | 32.13 | **36.09** | **65.70** | **55.11** | **61.59** |
| **Ours** | **20.22** | **26.52** | 45.13 | **54.38** | **51.44** | **32.85** | **36.09** | 64.26 | **56.93** | **61.76** |

Table 3: **CHAIR benchmark.** Lower scores (↓) on CHAIR$_S$, CHAIR$_I$ and higher (↑) F1 indicate better performance. The best results in each setting are **bolded**, and the second-best are underlined

| Method | LLaVA-1.5-7B | | | Qwen2.5-VL-7B | | |
|---|---|---|---|---|---|---|
| | CHAIR$_S$ ↓ | CHAIR$_I$ ↓ | F1 ↑ | CHAIR$_S$ ↓ | CHAIR$_I$ ↓ | F1 ↑ |
| Greedy | 53.8 | 14.66 | 82.33 | 35.2 | 8.43 | 79.85 |
| DoLA | 53.6 | 14.45 | 82.27 | 31.0 | 7.63 | 79.33 |
| VCD | 52.8 | 15.61 | 81.84 | 35.0 | 9.07 | 80.10 |
| M3ID | 57.0 | 16.57 | 82.06 | 33.8 | 9.80 | 78.80 |
| CODE | **45.2** | **13.13** | 81.48 | **17.0** | 9.35 | 48.50 |
| SID | 50.6 | 13.58 | **83.34** | 37.6 | 8.84 | 80.40 |
| **Ours** | 50.6 | 13.43 | 83.17 | 29.8 | **7.04** | **81.16** |

baselines, (c) *VCD* (Leng et al., 2024) mitigates hallucination by perturbing visual inputs with Gaussian noise and enforcing consistency between the original and perturbed outputs; (d) *M3ID* (Favero et al., 2024) contrasts outputs with and without visual input to alleviate language prior bias and enhance visual grounding; (e) *CODE* (Kim et al., 2024) leverages the model's self-generated image description as a contrasting reference to alleviate hallucination; (f) *SID* (Huo et al., 2025) applies contrastive decoding on low-attention vision patches to refine visual grounding.

**Implementation details.** In entire experiments, we apply deterministic **greedy** decoding (*i.e.*, temperature = 0, top-$p$ = 1, top-$k$ = 1, num_beams = 1) with a maximum output length of 512 tokens for *all decoding methods*. Regarding baselines, we follow the default hyperparameters provided by their original implementations. For ReVisiT, we set the hyperparameters as follows: for LLaVA-1.5-7B, we select vision tokens mostly from the last decoder layer ("last"). For Qwen2.5-VL-7B, we select vision tokens mostly from all even-numbered decoder layers ("all"). The APC threshold $\alpha$ is set for each task. More details are presented in Appendix A.4.

## 4.2 MAIN RESULTS

We evaluate both *generative* (caption faithfulness) and *discriminative* (object presence) grounding across five benchmarks to assess robustness and generalizability.

**HallusionBench** (Guan et al., 2024) probes image–context reasoning and entangled hallucinations with a LLM-based judge. We focus on the visual-dependent setting. Results are presented in Table 2, Across both LLaVA-1.5-7B and Qwen2.5-VL-7B, ReVisiT outperforms all baselines on the three representative metrics (*qAcc*, *fAcc*, and *aAcc*) among all methods. Notably, ReVisiT exceeds the second-best method by more than 5 points on *qAcc*, and reaches $1.75\times$ the score of vanilla greedy (from 11.55 to 20.22).

**CHAIR** (Rohrbach et al., 2018) measures hallucination rates in image captioning by comparing generated captions to ground-truth object annotations from MS COCO (Lin et al., 2014) dataset. We report sentence-level (*CHAIR$_S$*) and instance-level (*CHAIR$_I$*) hallucination scores on 500 sampled images, and additionally include *F1* as an per-sample aggregate caption-level score. Table 3 shows that ReVisiT reduces hallucination while preserving or improving coverage on both models, maintaining at least second-best for all metrics.

Table 4: **POPE benchmark**. Average for 3 datasets and 3 types. Higher scores (↑) on Accuracy, Precision, and F1 indicate better performance. The best results in each setting are **bolded**, and the second-best are underlined. Full results are presented in Table 11.

| Method | LLaVA-1.5-7B | | | Qwen2.5-VL-7B | | |
|---|---|---|---|---|---|---|
| | Accuracy ↑ | Precision ↑ | F1 ↑ | Accuracy ↑ | Precision ↑ | F1 ↑ |
| Greedy | 79.47 | 74.48 | 82.36 | 84.23 | **93.01** | 82.53 |
| DoLA | 79.58 | 74.67 | 82.41 | 81.36 | 92.36 | 78.71 |
| VCD | 77.74 | 72.51 | 80.97 | 84.43 | 92.48 | 82.91 |
| M3ID | 79.48 | 74.48 | 82.37 | 83.56 | 91.52 | 82.00 |
| CODE | 78.58 | 73.98 | 81.35 | 83.55 | 90.20 | 82.26 |
| SID | 80.40 | 75.94 | 82.91 | **85.78** | 92.27 | **84.72** |
| **Ours** | **81.80** | **78.03** | **83.45** | 84.53 | 93.00 | 82.93 |

Table 5: **MMMU and VQAv2**. For MMMU, we report *Accuracy* and *Subject Win/Tie vs. Greedy*; for VQAv2, we report *Accuracy*. Higher scores (↑) indicate better performance for all metrics. The best results in each setting are **bolded**, and the second-best are underlined

| Method | MMMU | | | | VQAv2 | |
|---|---|---|---|---|---|---|
| | LLaVA-1.5-7B | | Qwen2.5-VL-7B | | LLaVA-1.5-7B | Qwen2.5-VL-7B |
| | Accuracy ↑ | Subj. Win/Tie vs Greedy ↑ | Accuracy ↑ | Subj. Win/Tie vs Greedy ↑ | Accuracy ↑ | Accuracy ↑ |
| Greedy | 34.39 | - | 51.11 | - | 69.47 | 65.80 |
| DoLA | 34.29 | 28 | 50.15 | 18 | 69.87 | 60.60 |
| VCD | 33.51 | 19 | 51.13 | 21 | 65.93 | 66.07 |
| M3ID | 34.51 | **29** | 46.63 | 15 | 70.33 | 59.87 |
| CODE | 34.48 | 22 | 42.79 | 5 | 66.00 | 46.60 |
| SID | 33.10 | 16 | 49.75 | 16 | 70.33 | 67.13 |
| **Ours** | **35.13** | 27 | **51.18** | 23 | **70.40** | **67.60** |

**POPE** (Li et al., 2023b) evaluates object presence with binary questions across three datasets (MS COCO, A-OKVQA (Schwenk et al., 2022), and GQA (Hudson & Manning, 2019)) and three query types (random, popular, adversarial). We report *Accuracy*, *Precision*, and *F1*. As summarized in Table 4, ReVisiT ranks first or second in the overall averages across all datasets and query types, indicating robust gains in discriminative grounding.

**MMMU** (Yue et al., 2024) provides 30 subjects of VQA including both multiple-choice and open-ended settings. We report overall *Accuracy* and a per-subject *win/tie vs. greedy* rate to capture subject-wise gains over vanilla decoding. Results are presented in Table 5 (left). Across 30 subjects, ReVisiT attains the highest overall accuracy on both models while improving subject-wise robustness.

**VQAv2** (Goyal et al., 2017) measures VQA correctness under inter-annotator variability via consensus accuracy over 10 annotations. As shown in Table 5 (right), ReVisiT improves general VQA performance and attains the best accuracy on both models: LLaVA-1.5-7B (69.47 → **70.40**), Qwen2.5-VL-7B (65.80 → **67.60**), surpassing the strongest baselines (e.g., 70.33 and 67.13).

**Overall.** ReVisiT succeeds across *generative* and *discriminative* settings, under both *manually designed* and *LLM-aided* evaluations, and on both *hallucination-focused* and *general VQA* tasks—without training, architectural changes, external modules, or multi-pass inference—by referencing intrinsic vision-token semantics. Comprehensive per-benchmark breakdowns, metrics, and extended tables are presented in Appendix A.2 .

### 4.3 ADDITIONAL ANALYSES

**Inference speed improvement.** To evaluate the inference efficiency of ReVisiT compared to baseline decoding strategies, we measure the per-token computational time. All measurements are conducted on an NVIDIA RTX A6000 GPU with 48GB memory and an Intel(R) Xeon(R) Gold 5320 CPU running at 2.20 GHz. We perform 6 warmup iterations before measurement to eliminate initial overhead, and each of the 300 samples is decoded 3 times per sample during evaluation.

We report the average decoding time per generated token, measured in milliseconds (ms/token). Other evaluation configurations, including preprocessing pipelines and evaluation scripts, follow the official CHAIR benchmark settings. Exact numerical values are provided in Table 12.

As shown in Figure 6, ReVisiT matches the inference efficiency of vanilla greedy decoding, increasing time by only $+1.9\%$ on LLaVA-1.5-7B and $+0.6\%$ on Qwen2.5-VL-7B. By contrast, methods that require additional forward passes (e.g., VCD / M3ID / SID / CODE) *approach or exceed* $2\times$ the greedy runtime. Even DoLa, which only adds a projection on intermediate layers each step, incurs $+29.34\%$ and $+14.64\%$ overhead on LLaVA-1.5-7B and Qwen2.5-VL-7B, respectively. These results highlight the key advantage of ReVisiT: it enhances visual grounding with negligible computational overhead, unlike prior methods that significantly compromise inference efficiency.

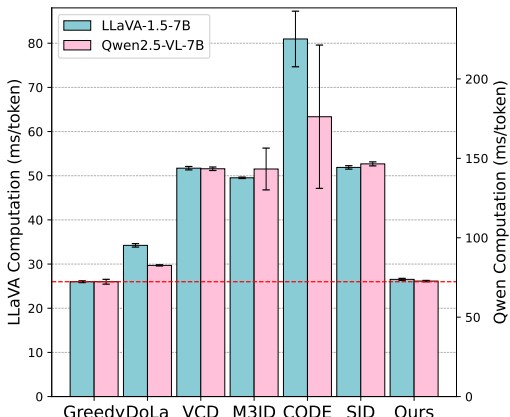

Figure 6: **Inference speed.** Comparison of per-token inference latency across different decoding strategies for LLaVA-1.5-7B (left y-axis) and Qwen2.5-VL-7B (right y-axis), with standard deviations visualized as error bars.

**Ablation study.** We ablate four factors on Qwen2.5-VL under the CHAIR benchmark (Figure 7): (1) vision-token selection criterion (min-/random/max-JSD), (2) vocabulary constraint (subset vs. full), (3) layer scope (all layers vs. last layer), and (4) APC threshold $\alpha$.

**Selection.** Under a constrained (subset) vocabulary, max-JSD collapses (F1 = 0.67) and random underperforms (F1 = 17.63), while our min-JSD attains the best performance (F1 = 81.16), substantiating the effectiveness of a context-aware constrained divergence for token selection (Table 10(a)).

**Vocabulary constraint.** Even with the same min-JSD selection, projecting over the full vocabulary fails (F1 = 1.52), exhibiting a degenerate behavior where vision-token projections are referenced off-target. This confirms that constraining the candidate set to context-relevant tokens is crucial (Table 10(a)); qualitative failures are provided in Appendix C.2.

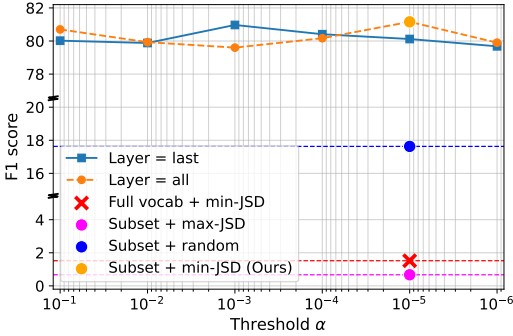

Figure 7: **Ablation study.** We evaluate the impact of vocabulary subset constraint, vision token selection, layer scope, and threshold $\alpha$ variation on CHAIR benchmark, reporting F1. Comprehensive results are presented in Table 10.

**Layer scope & threshold.** Across $\alpha$, the all-layers variant peaks at $\alpha = 10^{-5}$ (F1 = 81.16), while the last-layer variant peaks at $\alpha = 10^{-3}$ (F1 = 80.97). Overall sensitivity to $\alpha$ is modest (variations within $\approx 1.5$ F1 around each peak), indicating that *all-layers + constrained vocabulary + min-JSD* is a robust setting (Table 10b).

## 5 CONCLUSION

In this paper, we revisited the role of vision tokens in LVLM decoding and provided quantitative evidence that they encode *interpretable* semantics that become explicit under semantically coherent vocabulary constraints. Building on these findings, we introduced ReVisiT, a training-free, model-agnostic decoding strategy that references the most relevant vision token via a *context-aware constrained divergence* and refines token logits accordingly. As highlighted in Fig. 1, ReVisiT consistently improves performance across six model sizes spanning three architectures, demonstrating scalability and model-agnostic effectiveness.

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

# A    DETAILED DISCUSSIONS

## A.1    MOTIVATION

### A.1.1    METRIC FORMULATION (*at-least-one recall@*K)

Given any probability distribution $p_\theta(h, \mathcal{V})$ over the vocabulary $\mathcal{V}$,

let $\mathrm{TopK}_k(p_\theta) \subset \mathcal{V}$ denote the set of its top-$k$ tokens by probability, and let $G \subseteq \mathcal{V}$ be the set of ground-truth (GT) object synonyms relevant to the evaluation unit.

We define the success indicator

$$I(k) = \mathbf{1}\{ \mathrm{TopK}_k(q) \cap G \neq \varnothing \}, \tag{7}$$

and report the mean of $I(k)$ over all evaluation units in the set $\mathcal{U}$:

$$\widehat{\mathrm{Recall}}(k) = \frac{1}{|\mathcal{U}|} \sum_{u \in \mathcal{U}} I_u(k). \tag{8}$$

### A.1.2    FIGURE 3

CHAIR Rohrbach et al. (2018) identifies 275 hallucinated words across 500 captions. Among these, we focus on the subset where the hallucinated word and at least one ground-truth (GT) synonym are both representable as single tokens by the tokenizer, resulting in $n = 190$ steps available for step-level analysis. Table 6 reports the recall rates with 95% Wilson score intervals.

Table 6: At-least-one GT recall at hallucinated steps ($n = 190$) with 95% Wilson intervals.

| Top-$k$ | % of vocab | *at-least-one recall*@k [95% CI] |
|---|---|---|
| 50 | 0.033% | 63.2% [56.0, 70.0] |
| 100 | 0.066% | 73.7% [66.9, 79.6] |
| 500 | 0.33% | 95.8% [91.6, 98.1] |

### A.1.3    TABLE 1

**Setup formulation.**    We follow the notation in Sec. **??**: at decoding step $t$, the LVLM outputs $p_\theta(h_{T+t-1}^L, \mathcal{V})$ as in Eq. equation 1. For CHAIR, let $\Omega$ denote the MSCOCO synonym set ($|\Omega| = 403$). To ensure the clarity of object representation and avoid ambiguity from multi-token variants, we construct the single-token subset

$$\Omega^{(1)} = \{ w \in \Omega : \texttt{tokenize("\_w") has length 1} \}, \tag{9}$$

which contains 176 word types covering 62 object categories . At step $t$, we define the CHAIR-based ground-truth and hallucinated synonym subsets over $\Omega^{(1)}$ as $\mathcal{G}_t \subseteq \Omega^{(1)}$ and $\mathcal{H}_t = \Omega^{(1)} \setminus \mathcal{G}_t$. For step-level analyses, we further require that (i) the hallucinated word at $t$ is single-tokenizable and (ii) $\mathcal{G}_t \neq \varnothing$, yielding $n = 190$ paired steps (selection details match Appendix A.1.2). We evaluate two conditions: *w/ image* (vision tokens present) and *w/o image* (all vision tokens removed); in the latter, steps are aligned by teacher forcing with the previously generated caption while keeping all other decoding details identical.

**Statistical test and reporting.**    For paired step-level comparisons, we compute the maxima

$$m_t^{\mathrm{GT}} = \max_{w \in \mathcal{G}_t} p_\theta(w \mid h_{T+t-1}^L, \mathcal{V}), \qquad m_t^{\mathrm{Hal}} = \max_{w \in \mathcal{H}_t} p_\theta(w \mid h_{T+t-1}^L, \mathcal{V}),$$

and form

$$d_t^{\mathrm{with}} = m_t^{\mathrm{GT}} - m_t^{\mathrm{Hal}}, \qquad d_t^{\mathrm{w/o}} = m_t^{\mathrm{GT,w/o}} - m_t^{\mathrm{Hal,w/o}}, \qquad \Delta_t = d_t^{\mathrm{with}} - d_t^{\mathrm{w/o}}.$$

We apply the Wilcoxon signed-rank test (two-sided; Pratt) to $d_t^{\mathrm{with}}$, $d_t^{\mathrm{w/o}}$, and $\Delta_t$ on the intersection of steps available in both conditions. We report the paired sample size $N$, the median differences, and Holm-adjusted $p$-values for $\Delta_t$.

Table 7: Wilcoxon signed-rank results at the step level. $\Delta_t = (\mathrm{GT} - \mathrm{Hal})_{\mathrm{with}} - (\mathrm{GT} - \mathrm{Hal})_{\mathrm{w/o}}$. $N$ is the number of paired steps. Holm-adjusted $p$-values are shown for $\Delta_t$.

| Setting | $N$ | median($d^{\mathrm{with}}$) | median($d^{\mathrm{w/o}}$) | median($\Delta$) | $p_{\mathrm{Holm}}$ on $\Delta$ |
|---|---|---|---|---|---|
| All tokens | 88,706 | $-3.289$ | $-4.826$ | 0.715 | $< 10^{-300}$ |
| All tokens (in hallucinated set) | 35,116 | $-3.720$ | $-5.273$ | 0.774 | $1.09 \times 10^{-116}$ |
| Hallucinated steps | 190 | $-8.281$ | $-9.435$ | $-0.180$ | 0.95 |

In the full-token analyses, the median $\Delta$ is positive and highly significant, indicating a systematic shift of the step-level ordering toward GT when vision tokens are present. For the hallucinated-step subset ($n$=190), the median $\Delta$ is small and not significant, i.e., the reversed ordering (Hal > GT) persists at those positions despite the image.

### A.1.4 FIGURE 4

**Setup formulation.** We follow Eq. equation 4 with the final decoder layer ($j$=$L$). For each of the 500 MSCOCO validation images, we project every vision token's final-layer hidden state $h_i^L$ through the language-modeling head to obtain a distribution over the vocabulary,

$$q_i(w) \ = \ p_\theta(h_i^L, \mathcal{V}) \quad (i = 1, \ldots, |\mathbf{v}|),$$

yielding 173,173 vision-token projections in total. We evaluate two settings on the *same* projections: (i) full vocabulary $\mathcal{V}$ and (ii) a CHAIR-derived object subset $\Omega^{(1)}$ of 176 single-token synonyms covering 62 MSCOCO categories denoted as Eq. 9

**Main results with confidence intervals.** Table 8 summarizes token-level *at-least-one recall*@k with 95% Wilson intervals.

Table 8: At-least-one GT recall over $N$=173,173 vision-token projections with 95% Wilson intervals. GT targets are restricted to the same 62 categories in both settings.

| Top-$k$ | Full vocabulary | Constrained subset (176) |
|---|---|---|
| Top-1 | 2.03% [1.96, 2.10] | 40.44% [40.21, 40.67] |
| Top-5 | 7.87% [7.74, 8.00] | 69.56% [69.34, 69.78] |
| Top-10 | 12.38% [12.23, 12.54] | 79.78% [79.59, 79.97] |
| Top-30 | 21.72% [21.53, 21.91] | 89.24% [89.09, 89.39] |

### A.2 DATASET AND EVALUATION SETUP

**CHAIR evaluation.** CHAIR (Rohrbach et al., 2018) is a generative benchmark designed to measure object hallucinations in image captioning. A hallucinated object refers to any entity mentioned in the generated caption that is not present in the corresponding image. This metric is widely adopted to evaluate the visual grounding capability of vision-language models. We follow the standard CHAIR evaluation procedure based on the object annotations from the MS COCO dataset (Lin et al., 2014). Each image is associated with a set of ground-truth objects defined by the 80 category COCO detection vocabulary. During evaluation, noun phrases extracted from generated captions are matched against this predefined object list. Mentions of out-of-vocabulary objects are ignored, while in-vocabulary objects absent from the ground-truth annotations are considered hallucinated. We report two standard metrics provided by the CHAIR benchmark and one additional metric: We use the official CHAIR evaluation code[1] and randomly sample 500 images from the MS COCO validation set using a fixed seed (seed=42) for reproducibility. For each image, we prompt the model with "Please describe this image in detail." to generate captions.

---

[1] https://github.com/LisaAnne/Hallucination

**POPE evaluation.** POPE (Li et al., 2023b) is a discriminative benchmark designed to evaluate the object-level visual grounding capabilities of LVLMs. It formulates the task as a binary object presence question-answering problem, where the model is asked to determine whether a specific object is present in a given image. Evaluation is conducted using standard classification metrics derived from the confusion matrix, including accuracy, precision, and F1 score. Accuracy measures the overall proportion of correct predictions, while precision captures the fraction of predicted positive instances that are indeed correct. F1 score, defined as the harmonic mean of precision and recall, provides a balanced assessment of model performance on this binary classification task. The benchmark covers three datasets: MS COCO (Lin et al., 2014), A-OKVQA (Schwenk et al., 2022), and GQA (Hudson & Manning, 2019). For each dataset, POPE defines three types of query strategies—*random*, *popular*, and *adversarial*—yielding a total of nine evaluation scenarios. Binary questions are constructed in the form "Is there a <object> in the image?", with the queried object selected according to the designated strategy. Random queries are uniformly sampled from the dataset's object vocabulary. Popular queries focus on frequently occurring objects, while adversarial queries target objects that are semantically plausible in the given context but are absent from the image, thus probing the model's reliance on prior co-occurrence statistics rather than visual evidence. Ground-truth labels indicating object presence are provided by the official POPE benchmark[2]. For MS COCO, labels are derived from object detection annotations, whereas for A-OKVQA and GQA, they are obtained via SEEM-based segmentation annotations. Each query type consists of 3,000 binary QA instances, evenly split between positive and negative samples. As a result, we evaluate across a total of nine scenarios comprising 27,000 QA examples.

**LLaVA-Bench-In-the-Wild dataset.** We additionally employ the LLaVA-Bench-In-the-Wild (Liu et al., 2023) dataset to qualitatively evaluate the visual grounding ability of LVLMs in open-ended image understanding. The dataset consists of 24 diverse images, including both real-world photographs and abstract illustrations, and 60 textual prompts designed to elicit complex visual reasoning and language generation. Following prior work Leng et al. (2024), we present representative qualitative examples to illustrate how different decoding strategies affect grounding fidelity and response quality in challenging scenarios.

---

[2]https://github.com/AoiDragon/POPE

## A.3 RELATED WORKS

In this section, we provide a detailed discussion of prior works that aim to align visual inputs with language generation. We organize these approaches into three categories—*pre-alignment*, *intra-alignment*, and *post-alignment*—based on the stage where the alignment is applied with respect to the inference process. While this taxonomy follows the temporal flow of inference, the categories are largely orthogonal in methodology and can be composed to enhance multimodal grounding.

### A.3.1 PRE-ALIGNMENT

Recent advances in LVLMs have explored a range of *pre-alignment* strategies, in which visual and textual modalities are aligned during training to enhance multimodal understanding. Most existing approaches adopt a combination of instruction tuning and architectural refinements to achieve improved visual grounding. For instance, Qwen-VL Bai et al. (2023) and InternVL Chen et al. (2024c) retain decoder-only LLM backbones while incorporating high-resolution vision encoders and cross-modal projection modules. These models leverage large-scale curated image-text data and modality-specific tuning objectives, leading to strong general-purpose performance across a variety of vision-language tasks. Building upon these designs, Qwen2.5-VL Bai et al. (2025) introduces a Dynamic High-Resolution (DHR) processing pipeline and rotary position encodings tailored for vision-language alignment, further enhancing the model's ability to process diverse image resolutions efficiently. NVLM Dai et al. (2024) explores a broader architectural design space by implementing decoder-only, cross-attention, and hybrid variants, along with a DHR input pipeline and tile tagging mechanisms. On the other hand, there are another line of works designed to mitigate the architectural challenges within the pre-alignment space. Ferret You et al. (2024) focuses on fine-grained spatial referring and grounding by introducing hybrid region representations and a spatial-aware visual sampler, enabling the model to handle points, boxes, and free-form regions. In a different direction, Ovis Lu et al. (2024) addresses the structural mismatch between continuous visual embeddings and discrete textual embeddings. Specifically, Ovis introduces a learnable visual embedding table and a probabilistic vision tokenization scheme that mimics the structure of language token embeddings, thereby aligning the two modalities at the representation level. While these pre-alignment strategies have proven effective, they typically require extensive retraining and are not designed to enable inference-time intervention based on visual context.

### A.3.2 POST-ALIGNMENT

Post-alignment refers to strategies that operate on the generated output sequence, aiming to correct hallucinations or factual inconsistencies after the generation process is complete. Unlike pre- or intra-alignment methods that intervene during training or decoding, post-alignment introduces no changes to the model's parameters or inference-time logic. A key distinction among post-alignment methods lies in how the correction is applied. Some approaches train an auxiliary revision module, while others rely on external vision models to validate and refine the generated content. For example, LURE Zhou et al. (2023) learns a dedicated revisor that takes the original image and the LVLM-generated caption as input, and outputs a hallucination-corrected caption. The training signal is derived from statistical indicators such as co-occurrence bias and positional uncertainty observed in model outputs. In contrast, Woodpecker Yin et al. (2024) does not require finetuning, but instead performs post-hoc correction through a multi-stage pipeline. Starting from the generated text, it extracts visual concepts, formulates corresponding questions, and verifies them using a pretrained VQA model. Each stage of this process relies on existing pretrained components including question generation, visual verification, and text editing modules to systematically identify and correct hallucinated content. While post-alignment strategies offer a practical way to improve output consistency without altering the base model, they typically rely on additional training or auxiliary models.

### A.3.3 INTRA-ALIGNMENT

Improving visual grounding (*e.g.*, mitigating object hallucination) of LVLMs during decoding has evolved through distinct methodological paradigms, each addressing unique aspects of visual-textual misalignment. The research trajectory begins with Contrastive Decoding Li et al. (2023a) strategies, pioneered by VCD Leng et al. (2024), which introduced input distortions to contrast original and perturbed visual distributions, thereby reducing reliance on language priors within LVLMs. This

foundation expanded into instruction-aware variants such as ICD Wang et al. (2024), which amplifies alignment uncertainty through textual prompt disturbances, and M3ID Favero et al. (2024), which identifies language priors in the LLM decoder via vision-agnostic input perturbations. Concurrently, attention-centric approaches emerged Gong et al. (2024); Liu et al. (2024b); Woo et al. (2024); Huo et al. (2025); Kang et al. (2025), aiming to mitigate attention misallocation by recalibrating cross-modal attention or introducing contrastive signals for token-level grounding. Other paradigms Chen et al. (2024a); Huang et al. (2024); Wang et al. (2025) have also been explored in recent literature, but all these intra-alignment approaches require additional computation to obtain vision-related reference information.

---

**Algorithm 1** ReVisiT decoding algorithm

---

**Input:** LVLM $\mathcal{M}$ with final decoder layer $L$, projection (language modeling) head $W_{\texttt{proj}}$, and vocabulary $\mathcal{V}$), input image $I$, set of candidate decoder layers $\mathcal{J}$, input text prompt $x_{\text{raw}}$

---

Tokenize input text: $\mathbf{x} = \{x_0, \ldots, x_{|\mathbf{x}|-1}\} \leftarrow \texttt{Tokenizer}(x_{\text{raw}})$
Encode image: $\mathbf{v} = \{v_0, \ldots, v_{|\mathbf{v}|-1}\} \leftarrow \texttt{VisionEncoder}(I)$
Construct input sequence: $\mathbf{z} = \mathbf{v} \parallel \mathbf{x}$          $\triangleright \mathbf{z} = \{z_0, \ldots, z_{T-1}\}$ where $T = |\mathbf{v}| + |\mathbf{x}|$
Initialize output sequence $\mathbf{y} \leftarrow \emptyset$                       $\triangleright t \leftarrow 0$
**while** not end-of-sequence **do**
     Get decoder hidden states $h^L_{T+t-1}$ from $\mathcal{M}$ with context $(\mathbf{z}, \mathbf{y})$
     **if** first-timestep **then**
         Given $\{h^j_i\}$ for $i = 0, \ldots, T-1$ and $j \in \mathcal{J}$ from initial forward pass
         Initialize cache $\mathbf{l}_{\texttt{vision}} \leftarrow \emptyset$
         **for** each layer $j \in \mathcal{J}$ **do**
             **for** $i = 0$ to $|\mathbf{v}| - 1$ **do**
                 $l^j_i = W^\top_{\texttt{proj}} h^j_i$
                 Append $l^j_i$ to $\mathbf{l}_{\texttt{vision}}$
             **end for**
         **end for**
         Cache matrix $\mathbf{l}_{\texttt{vision}} \in \mathbb{R}^{|\mathcal{J}| \cdot |\mathbf{v}| \times |\mathcal{V}|}$
         first-timestep $\leftarrow$ False
     **end if**
     Compute vanilla distribution $p_{\text{base}} = \texttt{softmax}_{\mathcal{V}}(W^\top_{\texttt{proj}} h^L_{T+t-1})$ (Eq. 1)
     Select constrained vocabulary $\mathcal{V}^t_{\texttt{cons}} \subseteq \mathcal{V}$ based on APC of $p_{\text{base}}$ (Eq. 3)
     Let $p^{\text{cons}}_{\text{base}} = \texttt{softmax}_{\mathcal{V}^t_{\texttt{cons}}}(W^\top_{\texttt{proj}} h^L_{T+t-1})$
     $(j^*, i^*) \leftarrow \arg\min_{(j,i)} \texttt{JSD}(p^{\text{cons}}_{\text{base}} \| \texttt{softmax}_{\mathcal{V}^t_{\texttt{cons}}}(\mathbf{l}_{\texttt{vision}}[j, i]))$
     Let $p^{\text{cons}}_{\text{ref}} = \texttt{softmax}_{\mathcal{V}^t_{\texttt{cons}}}(\mathbf{l}_{\texttt{vision}}[j^*, i^*])$
     Logit adjustment via element-wise multiplication: $l_{\text{ReVisiT}} = p^{\text{cons}}_{\text{base}} \odot p^{\text{cons}}_{\text{ref}}$
     Normalize over constrained vocabulary: $p_{\text{ReVisiT}} = \texttt{softmax}_{\mathcal{V}^t_{\texttt{cons}}}(l_{\text{ReVisiT}})$
     Sample next token: $y_t \sim p_{\text{ReVisiT}}$
     Append $y_t$ to $\mathbf{y}$
     $t \leftarrow t + 1$
**end while**
**return** $\mathbf{y}$

---

## A.4 ReVisiT

In this section, we first provide the pseudo-code of ReVisiT in Algorithm 1, which illustrates how vision tokens are leveraged to refine the output distribution at each decoding step. We highlight that our implementation is memory-efficient, model-agnostic, and requires no additional training or modification to the base model; the decoding function alone is modified. ReVisiT involves two hyperparameters: (1) the pool of decoder layer(s) where vision tokens are selected, and (2) the alignment weight $\alpha$ that determines the size of constrained vocabulary subset. For LLaVA-1.5-7B, we empirically found that using vision tokens from the final decoder layer ("last") mostly yielded the best grounding fidelity. In contrast, for Qwen2.5-VL-7B, aggregating vision tokens from all even-numbered decoder layers ("all") was more effective. These selections were fixed per model across all benchmarks. The threshold $\alpha$ controls the size of the constrained vocabulary subset. Lower values (resulting in larger subsets) are used for generative tasks where broader linguistic diversity is desired, while higher values (yielding smaller subsets) are preferred in discriminative tasks to enhance precision. We select $\alpha$ from a logarithmic range between $10^{-1}$ and $10^{-6}$, with $10^{-5}$ serving as the default when task-specific adaptation is not required.

Table 9: Comprehensive results on CHAIR benchmark across various model sizes and architectures. Lower scores ($\downarrow$) on $CHAIR_S$, $CHAIR_I$ and higher ($\uparrow$) F1 indicate better performance. The best results in each model are **bolded**.

| Model | Method | $CHAIR_S \downarrow$ | $CHAIR_I \downarrow$ | F1 $\uparrow$ |
|---|---|---|---|---|
| LLaVA-1.5-7B | Greedy | 53.8 | 14.66 | 82.33 |
| (Liu et al., 2024a) | **Ours** | **50.6** | **13.43** | **83.17** |
| Qwen2.5-VL-7B | Greedy | 35.2 | 8.43 | 79.85 |
| (Bai et al., 2025) | **Ours** | **29.8** | **7.04** | **81.16** |
| InternVL2.5-8B | Greedy | 33.4 | 8.47 | 80.58 |
| (Chen et al., 2024b) | **Ours** | **29.8** | **7.81** | **81.93** |
| LLaVA-1.5-13B | Greedy | **48.0** | 13.29 | 83.27 |
| (Liu et al., 2024a) | **Ours** | 52.6 | **13.25** | **83.50** |
| InternVL3-14B | Greedy | 33.2 | 9.34 | 81.49 |
| (Zhu et al., 2025) | **Ours** | **33.0** | **8.66** | **82.03** |
| InternVL2.5-26B | Greedy | 31.8 | 8.39 | 79.60 |
| (Chen et al., 2024b) | **Ours** | **30.4** | **7.42** | **80.71** |
| Qwen2.5-VL-32B | Greedy | 53.6 | 8.85 | 82.00 |
| (Bai et al., 2025) | **Ours** | **53.4** | **8.5** | **82.55** |

# B  DETAILED QUANTITATIVE RESULTS

## B.1  COMPREHENSIVE RESULTS FOR 10 MODELS

## B.2  COMPREHENSIVE ABLATION RESULTS

Table 10: **Ablation study.** Panel (a) varies the vision token selection criterion (*min/max* = min-/max-JSD, *random* = uniform) and use of a vocabulary subset (*full* = no constraint, *subset* = APC). Panel (b) varies the layer scope (*last* vs *all*) and the APC threshold $\alpha$. Metrics are $CHAIR_S$ ($\downarrow$), $CHAIR_I$ ($\downarrow$), and F1 ($\uparrow$). Highlighted rows mark the main configuration.

| (a) Vocabulary Constraint $\times$ Token Selection | | | | |
|---|---|---|---|---|
| Vocab | Selection | $CHAIR_S \downarrow$ | $CHAIR_I \downarrow$ | F1 $\uparrow$ |
| subset | max | 0.0 | 0.00 | 0.67 |
| subset | random | 2.6 | 7.94 | 17.63 |
| subset | min | 29.8 | 7.04 | 81.16 |
| full | min | 0.2 | 7.14 | 1.52 |

| (b) Layer Scope $\times$ Threshold $\alpha$ | | | | |
|---|---|---|---|---|
| Layer Scope | $\alpha$ | $CHAIR_S \downarrow$ | $CHAIR_I \downarrow$ | F1 $\uparrow$ |
| last | 1e−1 | 33.8 | 8.32 | 80.02 |
| last | 1e−2 | 37.0 | 9.13 | 79.88 |
| last | 1e−3 | 34.0 | 7.88 | 80.97 |
| last | 1e−4 | 34.2 | 8.21 | 80.41 |
| last | 1e−5 | 31.4 | 8.04 | 80.12 |
| last | 1e−6 | 32.8 | 7.21 | 79.68 |
| all | 1e−1 | 33.8 | 8.82 | 80.70 |
| all | 1e−2 | 36.8 | 8.91 | 79.93 |
| all | 1e−3 | 38.8 | 9.36 | 79.60 |
| all | 1e−4 | 32.8 | 8.17 | 80.17 |
| all | 1e−5 | 29.8 | 7.04 | 81.16 |
| all | 1e−6 | 32.4 | 9.38 | 79.90 |

### B.3 COMPREHENSIVE POPE RESULTS

### B.4 TOKEN-LEVEL COMPUTATION SPEED

## C DETAILED QUALITATIVE RESULTS

### C.1 CASE STUDIES FOR VISION TOKEN ANALYSIS

To support the qualitative visualization presented in Figures 5 and 2, we provide the detailed numerical values and positional metadata used in our vision token analysis. All examples are drawn from the LLaVA-Bench-In-the-Wild dataset, using the LLaVA-1.5-7B. Vision token embeddings are extracted from decoder layer $L = 32$, and the threshold for constrained vocabulary selection is set to $\alpha = 10^{-5}$.

First, Figure 5 presents a case study from the LLaVA-Bench-In-the-Wild dataset (image ID: `002.jpg`, query: *"How many uncut fruits are in the image?"*). At decoding step $t = 2$, ReVisiT selects vision token index `229` as the most relevant reference based on the constrained vocabulary distribution. Table 13 reports the log-probability values from three sources: (1) the base model output (Eq. 1), (2) the vision token projection at index `229` (Eq. 4), and (3) the final combined distribution after log-probability fusion (Eq. 6). These values are computed over the constrained subset $\mathcal{V}^2_{\text{cons}}$. The token ``three'' is ultimately selected, as the vision token significantly boosts its relevance, increasing its probability from 35.45% to 58.27%, while suppressing ``four'' from 50.78% to 38.21%.

Next, Figure 2 visualizes a comprehensive case from the same dataset (image ID: `003.jpg`, prompt: *"Describe this photo in detail."*). Four vision tokens are selected to highlight different aspects of visual grounding. The three tokens shown on the **left** of the figure are used to examine interpretability and semantic alignment of vision token projections. For each token, we compute the projected distribution over (1) the full vocabulary $\mathcal{V}$ and (2) a manually defined constrained subset $\mathcal{V}_{\text{env}} =$ {river, mountain, person, sky, tree, cloud, sea, grass, rock}. This comparison reveals how visually grounded semantics become more prominent when irrelevant lexical candidates are excluded. The full distribution shifts are detailed in Table 14.

The token on the **right** (vision token index `193`) is selected to demonstrate the actual decoding behavior of ReVisiT. At generation step $t = 72$, this token serves acts as a reference informer to refine the output distribution through log-probability fusion. The final token ``painting'' is selected, with its probability increasing from 8.08% (base) to 39.15% (final), while alternatives such as hallucinatory token ``person'' are suppressed. Detailed values are presented in Table 15.

## C.2 WITHOUT VOCABULARY SUBSET CASE STUDY

**Q. Describe this photo in detail.**

*w/o subset*

Certainly!
This photographic depiction renders bustling aerospace窗 Cảnh Cảnh Cảnh Cảnh Cảnh Cảnh Cảnh Cảnh Cảnh Cảnh Cảnh Cảnh Cảnh Cảnh Cảnh Cảnh Cảnh Cảnh Cảnh Cảnh Cảnh Cảnh Cảnh Cảnh Cảnh Cảnh Cảnh Cảnh Cảnh Cảnh Cảnh Cảnh Cảnh Cảnh Cảnh Cảnh Cảnh Cảnh Cảnh Cảnh Cảnh Cảnh Cảnh Cảnh Cảnh Cảnh Cảnh Cảnh Cảnh Cảnh Cảnh Cảnh Cảnh Cảnh Cảnh Cảnh Cảnh Cảnh Cảnh Cảnh Cảnh Cảnh Cảnh Cảnh Cảnh Cảnh Cảnh Cảnh Cảnh Cảnh Cảnh Cảnh Cảnh Cảnh Cảnh Cảnh Cảnh Cảnh Cảnh Cảnh Cảnh Cảnh Cảnh Cảnh Cảnh Cảnh Cảnh Cảnh Cảnh ...

*w/ subset*

The image is a view from inside a terminal, looking out through a window, at an airport tarmac. The scene is overcast, with a cloudy sky, and the terminal's distinctive architecture, featuring a series of arched, white support structures, is visible in the background. A Thai Airways plane, identifiable by the purple and white livery and the Thai Airways logo, is parked at the gate, connected to the terminal. The aircraft's tail is prominently visible, with the Thai Airways logo and a gold and purple design.
In the foreground, a row of parked buses, likely for ground transportation, is visible. The buses are purple and white, and some of them have the Thai Airways logo. A few other vehicles, including a van and a small van, can be seen on the tarmac, and a few other aircraft and ground support equipment are in the distance. The overall setting is a busy airport terminal, with a focus on the Thai Airways plane and the terminal's unique architecture.

**Q. Describe this photo in detail.**

*w/o subset*

Certainly!
This captivating photographic portrayal brilliantly juxtaposes innocence—an modalità Cuộc Đè Ã Đè Ã Ã Ã Ã Ã Ã Ã Ã Ã Ã Ã Ã Ã Ã Ã Ã Ã Ã Ã Ã Ã Ã Ã Ã Ã Ã Ã Ã Ã Ã Ã Ã Ã Ã Ã Ã Ã Ã Ã Ã Ã Ã Ã Ã Ã Ã Ã Ã Ã Ã Ã Ã Ã Ã Ã Ã Ã Ã Ã Ã Ã Ã Ã Ã Ã Ã Ã Ã Ã Ã Ã Ã Ã Ã Ã Ã Ã Ã Ã Ã Ã Ã Ã Ã Ã Ã Ã Ã Ã Ã Ã Ã Ã Ã Ã Ã Ã Ã Ã Ã Ã Ã Ã Ã Ã Ã Ã Ã Ã Ã Ã Ã Ã Ã Ã Ã Ã Ã Ã Ã Ã Ã Ã Ã Ã Ã Ã Ã Ã Ã Ã Ã Ã Ã Ã Ã Ã Ã Ã Ã Ã Ã Ã Ã Ã Ã Ã Ã Ã Ã Ã Ã Ã Ã Ã Ã Ã Ã Ã Ã Ã Ã Ã Ã Ã Ã Ã Ã Ã Ã Ã Ã Ã Ã Ã Ã Ã Ã Ã Ã ...

*w/ subset*

This image shows the interior of a classic convertible car, likely a vintage model, with a black leather interior. The car's top is down, and the rear seats and dashboard are visible. The dashboard features a wood panel, a design characteristic of older car models, and the steering wheel is on the left side, indicating the car is likely from a country where driving on the left is the norm, such as the UK. The car's seats and door panels are upholstered in black leather, and the car's exterior is yellow, visible in the background. A dog with brown and black fur is sitting in the front passenger seat, looking out. The dog's expression appears calm and curious. The car's interior is clean and well-maintained, and the overall scene suggests a relaxed, leisurely moment, perhaps a car ride.

**Q. Describe this photo in detail.**

*w/o subset*

Certainly!
This image showcases neatly plated desserts arranged centrally upon plates atop what seems like dining seating—all盘겠겠겠겠겠겠겠겠겠겠겠겠겠겠겠겠겠겠겠겠겠겠겠겠겠겠겠겠겠겠겠겠겠겠겠겠겠겠겠겠겠겠겠겠겠겠겠 Ã Ã Ã Ã Ã Ã Ã Ã Ã Ã Ã Ã Ã Ã Ã Ã Ã Ã Ã Ã Ã Ã Ã Ã Ã Ã Ã Ã Ã Ã Ã Ã Ã Ã Ã Ã Ã Ã Ã Ã Ã Ã Ã Ã Ã Ã Ã Ã Ã Ã Ã Ã Ã Ã Ã Ã Ã Ã Ã Ã Ã Ã Ã Ã Ã Ã Ã Ã Ã Ã Ã Ã Ã Ã Ã Ã Ã Ã Ã Ã Ã Ã Ã Ã Ã Ã Ã Ã Ã Ã Ã Ã Ã Ã Ã Ã Ã Ã Ã Ã Ã Ã Ã Ã Ã Ã Ã Ã Ã Ã Ã Ã Ã Ã Ã Ã Ã Ã Ã Ã Ã Ã Ã Ã Ã Ã Ã Ã Ã Ã Ã Ã Ã Ã Ã Ã Ã Ã Ã Ã Ã Ã Ã Ã Ã Ã Ã Ã Ã Ã Ã Ã Ã Ã Ã Ã Ã Ã Ã Ã Ã Ã Ã ...

*w/ subset*

The image depicts a table with a central circular cutout, on which four plates of different colorful desserts are placed. The table is white, and the plates are black. The desserts are neatly arranged, each on its own plate, and include the following:
1. **Pink Roll Cake**: A rolled cake, possibly a Swiss roll, with a pink filling and a white layer, placed on a black plate with a knife resting on the side.
2. **Yellow Layer Cake**: A slice of cake with multiple yellow and cream-colored layers, placed on a black plate.
3. **Green Layer Cake**: A slice of cake with multiple green and cream-colored layers, placed on a black plate with a knife resting on the side.
4. **Patterned Roll Cake**: A rolled cake with a patterned surface, possibly a Swiss roll, with a yellow and white pattern, placed on a black plate with a knife resting on the side.
In the background, there is a stack of black plates, a chair with a patterned cushion, and a wooden floor. The setup suggests a dessert or tea time setup, with the table and plates arranged for serving.

Figure 8: **Without vocabulary subset case study.** Qualitative case study from Qwen2.5-VL-7B. `w/o subset` refers to ablation result of *without vocabulary subset constraint*, whereas `w/ subset` refers to our proposed ReVisiT.

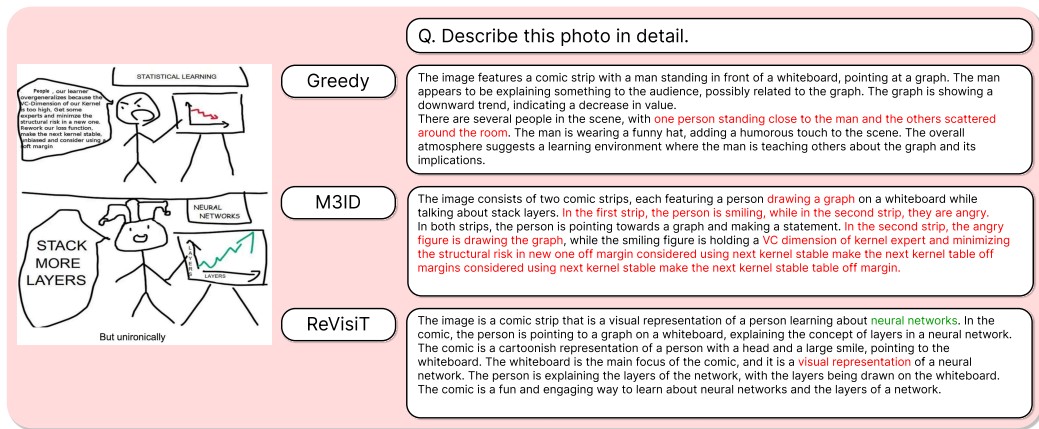

Figure 9: **Qualitative example.** The input image is a cartoon-style illustration contrasting classical statistical learning and neural network reasoning via a visual metaphor, emphasizing the shift from theoretical rigor to the heuristic of "stacking more layers." We compare the generated responses of vanilla greedy decoding, M3ID, and ReVisiT, highlighting how ReVisiT better captures the intended visual analogy compared to the baseline methods.

### C.3 QUALITATIVE EXAMPLES

To illustrate the qualitative improvements, we present examples from the LLaVA-Bench-In-The-Wild dataset, which consists of diverse real-world images requiring complex reasoning beyond conventional benchmarks. As shown in Figure 9, the input image is a cartoon illustration requiring contextual and conceptual understanding. While baseline methods such as vanilla greedy decoding fail to capture the main theme and generate overly generic or literal descriptions, or even getting stuck in repetitive phrasing. In contrast, ReVisiT successfully identifies the underlying concept: about neural network layers.

## D BROADER IMPACTS AND LIMITATIONS

**Broader impacts.** Like most approaches built upon LLMs, the proposed method inherits the limitations and potential societal risks of the underlying models, including biases in training data, reinforcement of stereotypes, and the generation of factually incorrect or harmful content. While our decoding strategy improves visual grounding by constraining generation based on vision token signals, it does not explicitly eliminate such biases and may inadvertently propagate them through vision-conditioned outputs. On the other hand, by incorporating grounded visual evidence during generation, our method may contribute to mitigating hallucinations and increasing the interpretability of multimodal outputs. We believe that this is an important step toward trustworthy and reliable deployment of LVLMs in the real-world.

**Limitations.** A potential limitation of our method is its strong reliance on the semantics encoded in vision tokens, which may lead to overfitting to visually salient cues. While this behavior improves visual grounding in most cases, it can result in an overemphasis on perceptually grounded elements at the expense of implicit, commonsense, or non-visual contextual information. To mitigate such overfitting issue, future work could explore adaptive interpolation strategies between reference logits and vanilla logits, or incorporate auxiliary confidence calibration mechanisms to balance multimodal importance dynamically. Another limitation of our approach is its exclusive reliance on the model's internal representations without access to external knowledge sources. As a result, it cannot correct factual errors or hallucinations originating from pretraining data, particularly in cases where visual evidence alone is insufficient for disambiguation or grounding. While our method improves the alignment between visual inputs and generated text, it does not incorporate retrieval mechanisms that could supplement the model with up-to-date or context-specific information. Nevertheless, due to its

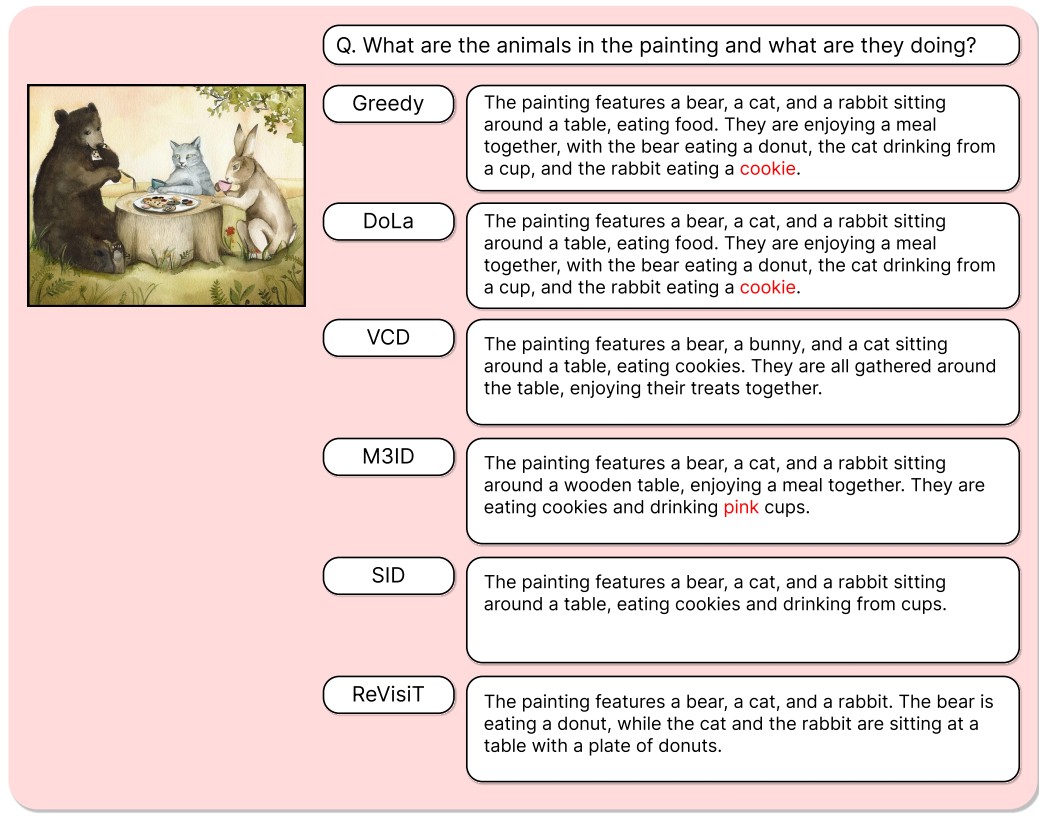

Figure 10: **Additional qualitative example.** The input image is a illustration showing a bear, a cat, and a rabbit seated around a table with a plate of donuts. We compare the responses of vanilla greedy decoding and ReVisiT to the question, *"What are the animals in the painting and what are they doing?"* While the greedy output introduces a hallucinated detail (\cookie") and assigns actions not visually supported (*e.g., "the cat drinking from a cup"*), ReVisiT provides a more faithful description aligned with the visual content, accurately identifying the animals and their activities.

modular and decoding-time nature, our method is compatible with retrieval-augmented generation (RAG) pipelines and could be integrated with external knowledge modules in future work.

Table 11: **Comprehensive results on POPE benchmark** across benchmarks (MS-COCO, A-OKVQA, GQA) and setups (Random / Popular / Adversarial). Higher scores (↑) on Accuracy, Precision, and F1 indicate better performance.

| | Setup | Method | LLaVA-1.5-7B | | | Qwen2.5-VL-7B | | |
|---|---|---|---|---|---|---|---|---|
| | | | Acc ↑ | Prec ↑ | F1 ↑ | Acc ↑ | Prec ↑ | F1 ↑ |
| MS-COCO | Random | Greedy | 89.07 | 89.54 | 89.00 | 84.07 | 99.71 | 81.09 |
| | | DoLa | 89.00 | 89.58 | 88.92 | 81.53 | **99.89** | 77.37 |
| | | VCD | 87.23 | 86.62 | 87.34 | 84.57 | 99.15 | 81.88 |
| | | M3ID | 89.07 | 89.54 | 89.00 | 83.93 | 99.32 | 80.96 |
| | | CODE | 86.53 | 87.23 | 86.41 | 84.00 | 98.02 | 81.26 |
| | | SID | **89.67** | **91.49** | **89.43** | **86.93** | 99.03 | **85.10** |
| | | **Ours** | 89.10 | 89.82 | 89.00 | 84.30 | 99.71 | 81.42 |
| MS-COCO | Popular | Greedy | 85.63 | 83.72 | 86.03 | 83.47 | 97.99 | 80.52 |
| | | DoLa | 85.63 | 83.85 | 86.00 | 81.10 | **98.54** | 76.96 |
| | | VCD | 83.53 | 80.75 | 84.25 | 83.93 | 97.39 | 81.27 |
| | | M3ID | 85.63 | 83.72 | 86.03 | 83.40 | 97.81 | 80.46 |
| | | CODE | 84.73 | 84.14 | 84.86 | 83.63 | 97.02 | 80.92 |
| | | SID | **86.13** | **85.19** | **86.32** | **86.23** | 97.22 | **84.42** |
| | | **Ours** | 85.80 | 84.16 | 86.13 | 83.70 | 98.01 | 80.85 |
| MS-COCO | Adversarial | Greedy | 79.27 | 74.72 | 81.01 | 83.10 | 97.06 | 80.16 |
| | | DoLa | 79.40 | 74.97 | 81.08 | 80.83 | **97.73** | 76.71 |
| | | VCD | 77.67 | 72.90 | 79.77 | 83.37 | 95.96 | 80.73 |
| | | M3ID | 79.27 | 74.72 | 81.01 | 83.00 | 96.79 | 80.06 |
| | | CODE | 78.67 | 75.23 | 80.02 | 83.00 | 95.41 | 80.31 |
| | | SID | **80.30** | **76.56** | **81.59** | **85.17** | 94.74 | **83.39** |
| | | **Ours** | 79.90 | 76.21 | 81.22 | 83.37 | 97.08 | 80.53 |
| A-OKVQA | Random | Greedy | 86.33 | 81.00 | 87.42 | 87.80 | 96.86 | 86.49 |
| | | DoLa | 86.43 | 81.25 | 87.47 | 85.33 | 96.99 | 83.26 |
| | | VCD | 83.70 | 77.91 | 85.23 | 87.87 | 96.56 | 86.62 |
| | | M3ID | 86.33 | 81.00 | 87.42 | 87.40 | 95.91 | 86.11 |
| | | CODE | 83.80 | 78.48 | 85.18 | 87.50 | 93.64 | 86.55 |
| | | SID | **87.57** | 83.48 | **88.28** | **88.83** | 96.12 | **87.88** |
| | | **Ours** | 87.37 | **84.75** | 87.83 | 88.27 | 96.90 | 87.08 |
| A-OKVQA | Popular | Greedy | 78.77 | 71.74 | 81.72 | 85.93 | 92.58 | 84.74 |
| | | DoLa | 79.00 | 72.06 | 81.85 | 83.37 | 92.17 | 81.43 |
| | | VCD | 76.47 | 69.58 | 79.99 | 85.90 | 92.10 | 84.78 |
| | | M3ID | 78.77 | 71.74 | 81.72 | 85.77 | 92.21 | 84.59 |
| | | CODE | 77.70 | 71.17 | 80.68 | **86.07** | 90.62 | **85.24** |
| | | SID | 80.63 | 74.30 | 82.87 | 87.00 | 92.11 | 86.16 |
| | | **Ours** | **81.97** | **77.54** | **83.31** | 86.30 | 92.44 | 85.23 |
| A-OKVQA | Adversarial | Greedy | 68.20 | 61.86 | 74.91 | 80.83 | 82.59 | 80.30 |
| | | DoLa | 68.33 | 62.00 | 74.95 | 79.63 | **84.22** | 78.17 |
| | | VCD | 67.80 | 61.69 | 74.47 | **81.10** | 82.69 | **80.63** |
| | | M3ID | 68.20 | 61.86 | 74.91 | 79.90 | 81.00 | 79.54 |
| | | CODE | 68.30 | 62.23 | 74.61 | 79.37 | 78.73 | 79.59 |
| | | SID | 71.33 | 64.75 | 76.57 | 82.40 | 83.38 | 82.14 |
| | | **Ours** | **72.83** | **67.00** | **76.81** | 81.33 | 82.82 | 80.90 |
| GQA | Random | Greedy | 85.97 | 79.72 | 87.30 | 87.53 | 97.15 | 86.12 |
| | | DoLa | 86.07 | 79.92 | 87.36 | 83.47 | 94.66 | 81.10 |
| | | VCD | 83.77 | 77.16 | 85.53 | 87.83 | 96.78 | 86.55 |
| | | M3ID | 86.00 | 79.74 | 87.33 | 87.43 | 96.14 | 86.12 |
| | | CODE | 84.87 | 78.58 | 86.37 | 87.07 | 94.48 | 85.89 |
| | | SID | 85.97 | 79.72 | 87.30 | **88.80** | 96.78 | **87.88** |
| | | **Ours** | **87.13** | **83.47** | **87.80** | 87.80 | 97.09 | 86.46 |
| GQA | Popular | Greedy | 73.80 | 66.38 | 78.64 | 83.80 | 88.82 | 82.68 |
| | | DoLa | 73.97 | 66.56 | 78.73 | 78.57 | 83.71 | 76.80 |
| | | VCD | 71.87 | 64.76 | 77.32 | 83.33 | 87.09 | 82.44 |
| | | M3ID | 73.83 | 66.39 | 78.67 | 80.67 | 82.39 | 80.14 |
| | | CODE | 73.90 | 66.60 | 78.60 | 81.47 | 83.29 | 80.95 |
| | | SID | 73.80 | 66.38 | 78.64 | **84.27** | 87.25 | **83.61** |
| | | **Ours** | **78.67** | **72.42** | **81.28** | 83.77 | 88.28 | 82.75 |
| GQA | Adversarial | Greedy | 68.17 | 61.60 | 75.19 | 81.50 | 84.36 | 80.70 |
| | | DoLa | 68.40 | 61.80 | 75.30 | 78.37 | 83.32 | 76.63 |
| | | VCD | 67.63 | 61.22 | 74.82 | 82.00 | 84.63 | 81.29 |
| | | M3ID | 68.20 | 61.62 | 75.22 | 80.50 | 82.11 | 80.00 |
| | | CODE | 68.70 | 62.12 | 75.39 | 79.87 | 80.56 | 79.64 |
| | | SID | 68.17 | 61.60 | 75.19 | 82.37 | 83.79 | **81.99** |
| | | **Ours** | **73.40** | **66.91** | **77.68** | 81.93 | **84.71** | 81.18 |

Table 12: **Per-token inference latency** (ms/token) of different decoding strategies. We report the mean ± standard deviation over 300 samples.

| Method | LLaVA-1.5-7B | Qwen2.5-VL-7B |
|--------|--------------|---------------|
| Greedy | **26.0 ± 0.2** | **72.3 ± 1.5** |
| DoLa | 34.2 ± 0.4 | 82.6 ± 0.4 |
| VCD | 51.7 ± 0.4 | 143.4 ± 1.1 |
| M3ID | 49.5 ± 0.2 | 143.3 ± 13.2 |
| CODE | 81.0 ± 6.3 | 176.2 ± 45.1 |
| SID | 51.9 ± 0.4 | 146.5 ± 1.3 |
| **Ours** | 26.5 ± 0.2 | 72.7 ± 0.4 |

| Rank | Token | Base Logit | | (229) Vision Token Logit | | Final Logit | |
|---|---|---|---|---|---|---|---|
| | | Log-Prob | Prob (%) | Log-Prob | Prob (%) | Log-Prob | Prob (%) |
| 1 | three | −1.04 | 35.45 | −1.47 | 22.98 | −2.51 | 58.27 |
| 2 | four | −0.68 | 50.78 | −2.25 | 10.52 | −2.93 | 38.21 |
| 3 | two | −4.57 | 1.04 | −1.19 | 30.44 | −5.76 | 2.26 |
| 4 | five | −2.54 | 7.91 | −4.45 | 1.16 | −6.99 | 0.66 |
| 5 | a | −4.33 | 1.31 | −3.08 | 4.58 | −7.42 | 0.43 |
| 6 | six | −3.94 | 1.94 | −5.15 | 0.58 | −9.10 | 0.08 |
| 7 | | −5.10 | 0.61 | −4.22 | 1.47 | −9.32 | 0.06 |
| 8 | several | −5.75 | 0.32 | −5.39 | 0.45 | −11.15 | 0.01 |
| 9 | seven | −5.79 | 0.31 | −6.12 | 0.22 | −11.90 | 0.00 |
| 10 | un | −8.65 | 0.02 | −3.80 | 2.24 | −12.45 | 0.00 |
| 11 | at | −8.28 | 0.03 | −4.77 | 0.85 | −13.05 | 0.00 |
| 12 | one | −10.05 | 0.00 | −3.69 | 2.50 | −13.74 | 0.00 |
| 13 | in | −11.15 | 0.00 | −2.99 | 5.01 | −14.14 | 0.00 |
| 14 | eight | −7.38 | 0.06 | −6.78 | 0.11 | −14.16 | 0.00 |
| 15 | f | −9.52 | 0.01 | −4.76 | 0.86 | −14.28 | 0.00 |
| 16 | multiple | −7.66 | 0.05 | −6.66 | 0.13 | −14.32 | 0.00 |
| 17 | only | −8.68 | 0.02 | −5.89 | 0.28 | −14.57 | 0.00 |
| 18 | nine | −7.71 | 0.04 | −6.97 | 0.09 | −14.68 | 0.00 |
| 19 | half | −8.62 | 0.02 | −6.22 | 0.20 | −14.85 | 0.00 |
| 20 | many | −8.70 | 0.02 | −6.21 | 0.20 | −14.91 | 0.00 |
| 21 | all | −10.38 | 0.00 | −4.85 | 0.78 | −15.23 | 0.00 |
| 22 | still | −11.11 | 0.00 | −4.15 | 1.57 | −15.26 | 0.00 |
| 23 | an | −10.33 | 0.00 | −5.10 | 0.61 | −15.43 | 0.00 |
| 24 | some | −10.05 | 0.00 | −5.40 | 0.45 | −15.46 | 0.00 |
| 25 | more | −11.15 | 0.00 | −4.68 | 0.92 | −15.83 | 0.00 |
| 26 | about | −10.05 | 0.00 | −5.90 | 0.27 | −15.95 | 0.00 |
| 27 | both | −10.74 | 0.00 | −5.23 | 0.53 | −15.98 | 0.00 |
| 28 | the | −11.98 | 0.00 | −4.07 | 1.71 | −16.05 | 0.00 |
| 29 | not | −11.58 | 0.00 | −4.59 | 1.02 | −16.16 | 0.00 |
| 30 | over | −11.99 | 0.00 | −4.25 | 1.42 | −16.25 | 0.00 |
| 31 | different | −11.93 | 0.00 | −4.73 | 0.88 | −16.66 | 0.00 |
| 32 | ten | −8.56 | 0.02 | −8.15 | 0.03 | −16.71 | 0.00 |
| 33 | just | −10.97 | 0.00 | −5.78 | 0.31 | −16.75 | 0.00 |
| 34 | fruit | −11.81 | 0.00 | −5.04 | 0.65 | −16.85 | 0.00 |
| 35 | as | −12.06 | 0.00 | −4.82 | 0.81 | −16.88 | 0.00 |
| 36 | small | −12.11 | 0.00 | −4.86 | 0.78 | −16.97 | 0.00 |
| 37 | various | −10.39 | 0.00 | −6.71 | 0.12 | −17.10 | 0.00 |
| 38 | also | −11.94 | 0.00 | −5.21 | 0.55 | −17.15 | 0.00 |
| 39 | very | −11.80 | 0.00 | −5.54 | 0.39 | −17.35 | 0.00 |
| 40 | around | −11.28 | 0.00 | −6.07 | 0.23 | −17.35 | 0.00 |
| 41 | numerous | −10.02 | 0.00 | −7.51 | 0.05 | −17.54 | 0.00 |
| 42 | no | −10.70 | 0.00 | −6.93 | 0.10 | −17.64 | 0.00 |
| 43 | total | −11.17 | 0.00 | −6.80 | 0.11 | −17.97 | 0.00 |
| 44 | currently | −11.17 | 0.00 | −6.89 | 0.10 | −18.06 | 0.00 |
| 45 | cut | −11.78 | 0.00 | −6.46 | 0.16 | −18.24 | 0.00 |
| 46 | quite | −11.62 | 0.00 | −6.71 | 0.12 | −18.33 | 0.00 |
| 47 | few | −11.91 | 0.00 | −6.64 | 0.13 | −18.55 | 0.00 |
| 48 | twelve | −10.07 | 0.00 | −8.53 | 0.02 | −18.60 | 0.00 |
| 49 | lots | −11.52 | 0.00 | −7.37 | 0.06 | −18.89 | 0.00 |
| 50 | th | −11.34 | 0.00 | −7.63 | 0.05 | −18.98 | 0.00 |
| 51 | actually | −11.84 | 0.00 | −7.40 | 0.06 | −19.24 | 0.00 |
| 52 | approximately | −11.05 | 0.00 | −8.45 | 0.02 | −19.50 | 0.00 |
| 53 | exactly | −12.14 | 0.00 | −7.48 | 0.06 | −19.62 | 0.00 |
| 54 | plenty | −11.78 | 0.00 | −7.95 | 0.04 | −19.73 | 0.00 |
| 55 | eleven | −10.82 | 0.00 | −9.25 | 0.01 | −20.07 | 0.00 |

Table 13: **Detailed numerical values for Figure 5.** Token scores over the constrained vocabulary $\mathcal{V}^2_{\mathrm{cons}}$ at decoding step $t = 2$ with ReVisiT are presented. The selected vision token (index 229) amplifies the relevance of \three", increasing its probability from 35.45% (base) to 58.27% (final), while suppressing \four" from 50.78% to 38.21%.

| Vision Token | Top (Pink) | | | | Middle (Yellow) | | | | Bottom (Cyan) | | | |
|---|---|---|---|---|---|---|---|---|---|---|---|---|
| Vocab | $\mathcal{V}$ | | $\mathcal{V}_{\text{env}}$ | | $\mathcal{V}$ | | $\mathcal{V}_{\text{env}}$ | | $\mathcal{V}$ | | $\mathcal{V}_{\text{env}}$ | |
| Token | Log-Prob | % | Log-Prob | % | Log-Prob | % | Log-Prob | % | Log-Prob | % | Log-Prob | % |
| river | $-14.74$ | 0.00 | $-6.16$ | 0.21 | $-3.61$ | 2.70 | $-0.36$ | 69.75 | $-14.58$ | 0.00 | $-5.02$ | 0.66 |
| mountain | $-14.07$ | 0.00 | $-5.48$ | 0.42 | $-6.47$ | 0.15 | $-3.22$ | 3.99 | $-15.02$ | 0.00 | $-5.46$ | 0.43 |
| person | $-13.39$ | 0.00 | $-4.80$ | 0.82 | $-12.27$ | 0.00 | $-9.01$ | 0.01 | $-9.86$ | 0.01 | $-0.30$ | 73.81 |
| sky | $-8.66$ | 0.02 | $-0.07$ | 93.44 | $-6.76$ | 0.12 | $-3.51$ | 2.99 | $-12.82$ | 0.00 | $-3.26$ | 3.83 |
| tree | $-12.06$ | 0.00 | $-3.48$ | 3.09 | $-11.71$ | 0.00 | $-8.46$ | 0.02 | $-12.47$ | 0.00 | $-2.92$ | 5.42 |
| cloud | $-13.88$ | 0.00 | $-5.30$ | 0.50 | $-9.52$ | 0.01 | $-6.26$ | 0.19 | $-12.46$ | 0.00 | $-2.91$ | 5.47 |
| sea | $-13.80$ | 0.00 | $-5.21$ | 0.55 | $-6.82$ | 0.11 | $-3.57$ | 2.81 | $-13.24$ | 0.00 | $-3.68$ | 2.52 |
| grass | $-15.41$ | 0.00 | $-6.82$ | 0.11 | $-7.19$ | 0.08 | $-3.94$ | 1.95 | $-13.41$ | 0.00 | $-3.86$ | 2.11 |
| rock | $-13.35$ | 0.00 | $-4.77$ | 0.85 | $-4.95$ | 0.71 | $-1.70$ | 18.33 | $-12.41$ | 0.00 | $-2.85$ | 5.77 |

Table 14: **Detailed numerical values for Figure 2 (left).** Log-probabilities and probabilities (%) of selected constrained vocabulary tokens as projected by three vision tokens (pink, yellow, cyan) under the full vocabulary $\mathcal{V}$ and the constrained vocabulary $\mathcal{V}_{\text{env}}$.

| Rank | Token | Base Logit | | (193) Vision Token Logit | | Final Logit | |
|---|---|---|---|---|---|---|---|
| | | Log-Prob | Prob (%) | Log-Prob | Prob (%) | Log-Prob | Prob (%) |
| 1 | painting | $-2.52$ | 8.08 | $-1.27$ | 27.99 | $-3.79$ | 39.15 |
| 2 | rock | $-2.60$ | 7.42 | $-2.28$ | 10.22 | $-4.88$ | 13.11 |
| 3 | mountain | $-2.14$ | 11.76 | $-3.03$ | 4.83 | $-5.17$ | 9.82 |
| 4 | person | $-1.89$ | 15.10 | $-3.59$ | 2.75 | $-5.48$ | 7.19 |
| 5 | landscape | $-2.27$ | 10.30 | $-3.42$ | 3.27 | $-5.70$ | 5.82 |
| 6 | d | $-3.45$ | 3.16 | $-2.29$ | 10.14 | $-5.74$ | 5.55 |
| 7 | p | $-3.91$ | 2.01 | $-2.45$ | 8.60 | $-6.36$ | 3.00 |
| 8 | scene | $-3.80$ | 2.23 | $-2.70$ | 6.75 | $-6.50$ | 2.60 |
| 9 | small | $-2.83$ | 5.91 | $-3.70$ | 2.48 | $-6.52$ | 2.54 |
| 10 | beautiful | $-3.67$ | 2.54 | $-3.03$ | 4.84 | $-6.70$ | 2.13 |
| 11 | large | $-2.93$ | 5.34 | $-3.78$ | 2.28 | $-6.71$ | 2.11 |
| 12 | scen | $-3.57$ | 2.81 | $-3.17$ | 4.19 | $-6.74$ | 2.04 |
| 13 | hill | $-4.01$ | 1.82 | $-2.83$ | 5.91 | $-6.84$ | 1.86 |
| 14 | chair | $-2.85$ | 5.78 | $-4.38$ | 1.26 | $-7.23$ | 1.26 |
| 15 | bow | $-3.87$ | 2.09 | $-3.68$ | 2.53 | $-7.54$ | 0.92 |
| 16 | second | $-3.80$ | 2.23 | $-4.52$ | 1.09 | $-8.33$ | 0.42 |
| 17 | smaller | $-3.02$ | 4.90 | $-6.04$ | 0.24 | $-9.06$ | 0.20 |
| 18 | boat | $-3.00$ | 4.98 | $-6.34$ | 0.18 | $-9.34$ | 0.15 |
| 19 | distant | $-4.14$ | 1.59 | $-5.43$ | 0.44 | $-9.57$ | 0.12 |

Table 15: **Detailed numerical values for Figure 2 (right).** Token scores over the constrained vocabulary $\mathcal{V}_{\text{cons}}^{72}$ at decoding step $t = 72$ with ReVisiT. The selected vision token (index 193) amplifies the relevance of \painting", increasing its probability from 8.08% (base) to 39.15% (final) while suppressing hallucinatory token \person" from 15.10% to 7.19%.

