# OpenReview forum: "Revisit What You See: Disclose Language Prior in Vision Tokens for LVLM Decoding"
_ICLR.cc/2026/Conference — ICLR 2026 Conference Withdrawn Submission_

### Official Review · Reviewer_eYZ4 · 2025-10-18

**Soundness:** 2
**Presentation:** 2
**Contribution:** 2
**Rating:** 2
**Confidence:** 5

**Summary:**

This paper explores how visual information contributes to the decoding process of Large Vision-Language Models (LVLMs). Based on detailed analyses, it introduces ReVisiT, a simple yet effective training-free decoding method that dynamically selects the most relevant vision token at each decoding step via context-aware constrained divergence minimization, and refines the output probability distribution through constrained projection. Extensive experiments across five benchmarks demonstrate that ReVisiT consistently reduces hallucination with minimal computational overhead.

**Strengths:**

- The paper provides a thorough investigation into how vision tokens influence the decoding behavior of LVLMs, offering new insights into their internal visual grounding mechanisms.
- It proposes a training-free, efficient decoding strategy that effectively mitigates hallucination while preserving inference speed.
- The method is validated across multiple benchmarks and architectures, showing consistent improvements in both hallucination reduction and general visual-language performance.

**Weaknesses:**

- **Similarity to prior techniques**: The proposed adaptive context-aware vocabulary subset construction appears conceptually similar to the adaptive token truncation strategies introduced in prior works [1, 2, 5], raising concerns about novelty in this component.

- **Single-token selection limitation**: In Section 3.2, the method considers only the most relevant vision token at each decoding step. However, in many cases, multiple visual tokens jointly contribute to the next-token prediction. For instance, in Figure 5, several vision tokens depicting “uncut fruits” likely influence the generation of the token “three.” This simplification makes the approach somewhat unclear. It is also recommended to include visualizations of the selected vision token and its corresponding text token generation process to enhance interpretability and transparency.
- **Incomplete baseline comparison**: Several recent hallucination mitigation approaches, such as VTI [3], VASparse [4], and CMI-VLD [5], are not included in the comparison. These methods respectively address latent-space steering, vision-aware decoding, and adaptive cross-modal consistency. Including them would strengthen the empirical validation and contextual positioning of this work.
- **Limited model coverage**: The experiments are conducted only on LLaVA-1.5 and Qwen2.5-VL, while other representative LVLMs, such as LLaVA-NEXT and InstructBLIP, are not evaluated. Broader model coverage would better demonstrate generality.
- **Restricted benchmark scope**: The evaluation could be more comprehensive by incorporating  more benchmarks, such as MME [6], MMBench [7], or the GPT-4-assisted hallucination benchmark [8], to further assess robustness and generalization.
- **Typos and inconsistencies**: The manuscript contains several typographical and structural issues that affect readability. For example, Appendix B is missing; “FIGURE 3” appears incorrectly as a section title; references in Appendix A.1.3 are incomplete; and redundant expressions such as “Eq” and “equation” occur in Appendix A.1.4. These suggest a need for more careful proofreading.
- **Hyperparameter sensitivity**: The proposed method involves several hyperparameters (e.g., the layer scope, threshold $\alpha$) and requires model-specific tuning. This dependence may limit generalizability and increase the cost of deployment across different LVLMs.
- **Scalability concern**: The performance gains of the proposed method appear to diminish as the model size increases (e.g., Qwen2.5-VL-7B vs. Qwen2.5-VL-32B in Table 9), suggesting limited scalability to larger LVLMs.


[1]Leng, Sicong, et al. Mitigating object hallucinations in large vision-language models through visual contrastive decoding. In CVPR 2024.

[2]Huo Fushuo, et al. Self-introspective decoding: Alleviating hallucinations for large vision-language models. In ICLR 2025.

[3]Liu Sheng, et al. Reducing hallucinations in vision-language models via latent space steering. In ICLR 2025

[4]Zhuang Xianwei, et al. Vasparse: Towards efficient visual hallucination mitigation for large vision-language model via visual-aware sparsification. In CVPR 2025

[5]Fang Hao, et al. Grounding Language with Vision: A Conditional Mutual Information Calibrated Decoding Strategy for Reducing Hallucinations in LVLMs. In NIPS 2025.

[6]Cui, Can, et al. A survey on multimodal large language models for autonomous driving. In WACV 2024.

[7]Liu Yuan, et al. Mmbench: Is your multi-modal model an all-around player? In ECCV 2024.

[8]Zhao, Zhiyuan, et al. Beyond hallucinations: Enhancing lvlms through hallucination-aware direct preference optimization. arXiv preprint arXiv:2311.16839 (2023).

**Questions:**

Refer to Weaknesses.

---

### Official Review · Reviewer_usmk · 2025-10-22

**Soundness:** 3
**Presentation:** 2
**Contribution:** 2
**Rating:** 2
**Confidence:** 4

**Summary:**

The paper presents ReVisiT, a training-free methodology for VLMs that leverages the vision tokens to better guide the textual outputs. Compared to greedy decoding, the performance is structurally better, within 1 percent, across a series of VLMs.

**Strengths:**

To use a constrained vocabulary subset, which is more visually oriented and grounded, for text generation is interesting.

The insights in prevalence of groundtruth objects in top-probability predictions is interesting too.

**Weaknesses:**

Figure 2 does not help me to understand how the method technically comes to the proposed constrained V. I can see what it does, but not how it is done, on which rationale/principles, etc.

To me it appears that the constrained V is the key ingredient of the method. When it is defined (Eq 3) the paper states 'following Li et al'; is Eq 3 a repetition of another paper? If so, it seems that the methodological contribution is very limited.

The explaination of what Vcons in Eq 3 entails, what it does in the subsequent steps, is lacking. This is an essential step but it is not elaborated what Eq 3 means.

Related work is missing and a discussion of the limitations of the method is lacking.

**Questions:**

Is Eq 3 novel or adopted from Li et al?

Can you explain Eq 3 and what it entails and will do in subsequent steps of the model?

---

### Official Review · Reviewer_1BX1 · 2025-10-29

**Soundness:** 2
**Presentation:** 3
**Contribution:** 2
**Rating:** 4
**Confidence:** 3

**Summary:**

The paper, Revisit What You See, addresses the problem of visual hallucination in Large Vision–Language Models (LVLMs). The paper observes that while hallucinations occur, the vision tokens in these models still contain correct and meaningful visual information, and those information can be utilized to correct the decoding process. Authjors propose ReVisiT: at each decoding step, ReVisiT dynamically narrows the vocabulary to contextually relevant tokens, selects the vision token most aligned with the current decoding context via minimum JSD, and refines the model’s output logits by combining them with this selected vision token’s projected distribution. Experiment results demonstrate that ReVisiT in efficient and can sometimes improve performance over baseline decoding methods.

**Strengths:**

1. The overall idea of revisiting visual tokens for guidance makes sense to me. Shrinking and refining the decoding space based on visual tokens is a good external method to reduce hallucination.
2. Experiments are clear and comprehensive across models. Authors compare on LLaVA-1.5, Qwen2.5-VL, with various decoding methods, making the results reliable.
3. Computation overhead is negligible. This is a great advantage over other methods like SID.

**Weaknesses:**

1. The main concern stems from performance. On LLaVA-1.5, the performance growth is tangible; however, on the relatively newer model Qwen2.5-VL, the advantage over traditional "Greedy" decoding is not impressive (~0.2-0.3% in average). This points to a crucial question: Is this innovation still useful in modern or future models? (I will state this in the Question section)
2. Though the paper is generally well written, the abstract and Figure 2 are not expressive enough. Also, the example in Figure 2 is confusing, as "in the background, there is a painting of ..." does not correctly describe the picture.
3. Current benchmarks are focused on hallucination. To further improve the quality of paper, maybe more experiments on different datasets should be conducted to prove the effectiveness on general tasks, besides the HallusionBench, POPE, MMMU, VQAv2.

**Questions:**

I would appreciate your explanation on: What is the fundamental difference of your method on decoding with the Attentions in the VLM decoder layer, which can both promote interaction between language and visual tokens? Theoretically, the Attentions are more flexible and your method manually narrow down the search space, so when scaled up, well-trained Attentions should perform no less than your design. Can your method still bring significant benefit to the mainstream research in VLMs?

In my perspective, this is also the reason why your method's advantage shrink when carried to the better-trained Qwen2.5-VL from LLaVA-1.5.

---

### Note · Authors · 2025-11-28

**Comment:**

We thank the reviewers and the AC for their time and thoughtful feedback.

After reading the reviews, we realized that the current submission does not clearly convey the main contributions we intended to emphasize, particularly regarding the interpretability and generalizability of our method.

For this reason, we have decided to withdraw the paper and will carefully incorporate the reviewers' suggestions into a substantially revised future version of this work.

**Withdrawal Confirmation:**

I have read and agree with the venue's withdrawal policy on behalf of myself and my co-authors.